# Defining the genetic and evolutionary architecture of alternative splicing in response to infection

Maxime Rotival[1], Hélène Quach[1] & Lluis Quintana-Murci [1]

Host and environmental factors contribute to variation in human immune responses, yet the genetic and evolutionary drivers of alternative splicing in response to infection remain largely uncharacterised. Leveraging 970 RNA-sequencing profiles of resting and stimulated monocytes from 200 individuals of African- and European-descent, we show that immune activation elicits a marked remodelling of the isoform repertoire, while increasing the levels of erroneous splicing. We identify 1,464 loci associated with variation in isoform usage (sQTLs), 9% of them being stimulation-specific, which are enriched in disease-related loci. Furthermore, we detect a longstanding increased plasticity of immune gene splicing, and show that positive selection and Neanderthal introgression have both contributed to diversify the splicing landscape of human populations. Together, these findings suggest that differential isoform usage has been an important substrate of innovation in the long-term evolution of immune responses and a more recent vehicle of population local adaptation.

[1] Human Evolutionary Genetics Unit, Institut Pasteur, CNRS UMR2000, 25-28 rue Dr Roux, Paris 75015, France. Correspondence and requests for materials should be addressed to M.R. (email: maxime.rotival@pasteur.fr) or to L.Q.-M. (email: quintana@pasteur.fr)

Alternative splicing (AS) is an essential mechanism for generating functional diversity, as it allows individual genes to express multiple mRNAs and encode numerous proteins, through rearrangement of existing domains[1]. It is estimated that nearly 95% of mammalian genes undergo AS, with strong impact on essential regulatory processes such as chromatin modification and signal transduction[1,2]. Because it allows to increase protein diversity at a minimal cost for the organism, AS has undergone rapid evolution across vertebrates[3–5] and is thought to play a key role in primate adaptation, including humans[6,7]. One function where innovation is essential is immunity, as the constant arms race against invading pathogens requires the ability of the host to rapidly adapt to new pathogenic cues, while maintaining homeostasis[8–10]. Thus, if we are to fully understand the regulation of immune function in humans, we need to dissect the degree of population-level variation of AS and define its genetic and evolutionary determinants in a cellular setting relevant to immunity to infection.

Previous studies, fuelled by the advent of RNA sequencing, have defined the splicing landscape of a large variety of human cells and tissues[11–14]. The contribution of AS in the context of immunity is increasingly recognised[13–16], and a few recent studies have reported widespread post-transcriptional modifications in response to environmental cues including infection[17–21]. However, there is increasing evidence to suggest that a large fraction of isoform diversity results from the usage of cryptic splice sites, leading to the formation of non-functional transcripts, a phenomenon known as noisy splicing[22–24]. Yet, the degree of such a stochastic noise in immune cells and how cellular perturbation with external stimuli, such as infection, generates non-functional transcripts remain to be determined.

The genetic determinants of AS are increasingly well-defined by studies of splicing quantitative trait loci (sQTLs)[19,20,25–32] and massively parallel splicing assays[33,34], which indicate that splicing is under strong genetic control in humans and, in most cases, has direct effects on protein sequences. Furthermore, besides the known role of splice-altering variants in Mendelian disorders[35,36], there is increasing support to the notion that they can also contribute to common disease risk, broadening our understanding of the mechanistic links between genetic variation, gene regulation and ultimate phenotypes[19,20,27,35,37]. However, the genetic determinants of differential isoform usage in response to immune stimulation, which inform gene–environment (G × E) interactions, and their contribution to complex immune-related traits are far from clear.

Recent studies have also shown that the intensity of immune responses, defined through transcriptional profiling of macrophages and monocytes exposed to various bacterial and viral challenges, differs substantially across human populations, owing to past adaptation to pathogen pressure[29,38]. Similarly, variation in AS has been reported across populations from different ethnic backgrounds[28,39,40], and signals of selection have been detected in splicing regulatory elements[7,41]. Yet, the contribution of differential isoform usage to ancestry-related differences in immune responses and the role of splicing as a vehicle of recent population adaptation, through different evolutionary mechanisms, are largely unknown.

In this study, we leverage RNA-sequencing data of human primary monocytes both at the basal state and after stimulation with different ligands, in 200 healthy individuals of African and European descent. We characterise the splicing landscape of the innate immune response and explore both long-term and recent evolution of immune gene splicing. We show that immune stimulation has a pervasive impact on AS, leading to increased isoform diversity but also elevated levels of noisy splicing. We map the genetic determinants of AS, including those that manifest in a context-specific manner, and uncover their contribution to chronic immune-related disorders. Finally, using population genetics tools, we show that positive selection and admixture with Neanderthals have both contributed to shape the current population variability in the splicing landscape of the human immune response.

## Results

**Characterising the landscape of splicing in human monocytes.** To characterise the diversity of splicing in immune responses, we extracted ~7.2 billion spliced reads from 970 RNA-sequencing profiles obtained from resting and activated monocytes, originating from 200 individuals of African and European ancestry[38]. Monocytes were activated with various Toll-like receptor (TLR) ligands lipopolysaccharides (LPS), $Pam_3CSK_4$ and R848 activating TLR4, TLR1/2 and TLR7/8 pathways, respectively), or a live strain of influenza A virus (IAV). After excluding reads that did not match known exon–exon junctions (<0.4% of all spliced reads, Supplementary Fig. 1a–c), we identified ~1.9 million unique junctions, defining over 1.7 million acceptor and donor sites. Among the 1.1 million splice sites that could be unambiguously assigned to a gene expressed with more than 1 fragment per kilobase per million (FPKM > 1), 80% presented weak activity across conditions (i.e., <1 supporting read per sample or the reads supporting the splice site account for <5% of all reads, referred as weak splice sites). The remaining 20%, corresponding to 217,792 high-activity splice sites, were composed of 56,049 sites that were active in a subset of transcripts (i.e., reads supporting the splice site account for 5–95% of all reads, alternative splice sites), and 161,743 sites that were constitutively active in our dataset (>95% of supporting reads, constitutive splice site). These observations, which set the bases of our study, indicate pervasive, AS in resting and activated monocytes.

**Reduced long-term conservation of immune gene splicing.** We first investigated the evolutionary pressures characterising the various types of splice sites (weak, alternative and constitutive) in the whole dataset, using the Genomic Evolutionary Rate Profiling (Gerp) statistical framework[42]. We observed a bimodal distribution of Gerp Rejected Substitutions (RS) scores (Fig. 1a), with 60% of splice sites showing little to no conservation (GerpRS < 2). The percentage of conserved sites increased with splice site usage, across all levels of gene expression (Supplementary Fig. 1d–f). Thus, 98 and 71% of constitutive and alternative splice sites, respectively, showed moderate to high conservation (GerpRS > 2), whereas only 25% of weak splice sites were conserved (Fig. 1b). Among weak splice sites that were not conserved, 95% were absent from Ensembl annotations and were thus considered as cryptic splice sites. Interestingly, we found that active, non-conserved splice sites are preferentially observed in genes with functions related to immune response (likelihood-ratio test; odds ratio (OR) > 1.2, $p < 3.2 \times 10^{-11}$, Supplementary Data 1). Furthermore, we found that genes whose expression was induced upon stimulation in our setting ($\log_2 FC > 1$) were also more likely to harbour non-conserved splice sites, relative to non-induced genes that were expressed at similar levels, with the strongest enrichment being found in response to IAV stimulation (likelihood-ratio test; OR > 2.3, $p < 4.2 \times 10^{-24}$, Supplementary Data 1).

We next determined whether such a decreased conservation reflects higher redundancy of ancient splice sites or increased rate of recent splice sites among immune genes, and dated human splice sites based on their occurrence in the vertebrate phylogeny (Supplementary Note 1). Immune genes were defined based on both Gene Ontology (GO:0006955, immune response) and their

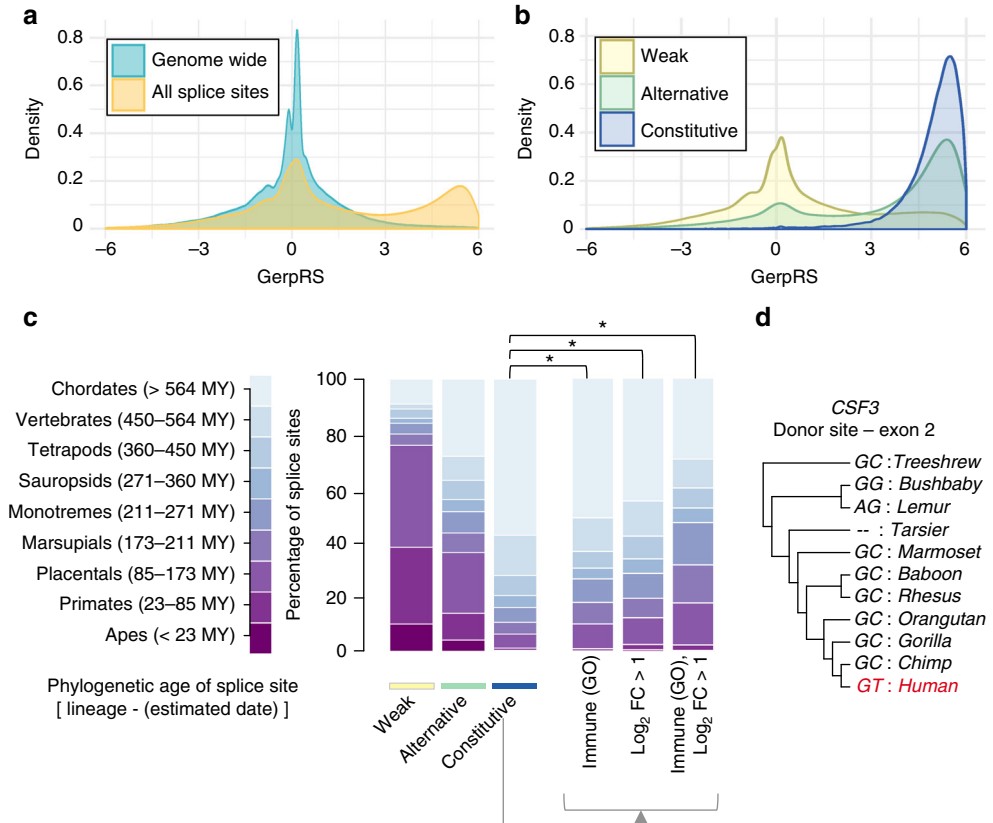

**Fig. 1** Splice sites of immune genes show decreased inter-species conservation. **a** Conservation of splice sites shows a strong bimodal distribution (orange). One-third of splice sites show strong conservation (GerpRS > 2), whereas the remaining splice sites match genome-wide expectations (turquoise). **b** Breakdown of GerpRS scores according to splice site activity (weak, alternative or constitutive). **c** Phylogenetic age of donor/acceptor sites according to splice site activity (weak, alternative or constitutive), and breakdown of the age of constitutive splice sites, across various categories of genes. These include all genes, immune response genes (GO:0006955), stimulation-induced genes and the intersection of the latter two categories. Significance was obtained by resampling, matching for the phylogenetic age of the gene and its expression. *Resampling $p < 0.001$. **d** Example of a human-specific splice site at the *CSF3* gene. The phylogenetic tree is drawn according to the established phylogeny, and aligned sequences are shown at the two-base pairs matching the human splice site. Dashes (–) indicate alignment gap. Source data are provided as a Source Data file

fold-change of expression in response to stimulation ($\log_2$FC > 1 in at least one stimulation condition). Overall, the age of splice sites was found to be strongly correlated to their activity; for instance, 94% of constitutive splice sites appeared before the split with marsupials more than 173 MY ago (Fig. 1c). We then assessed the age of splice sites of immune genes, and found that their constitutive splice sites were 8–12% younger than those of random genes with similar expression levels and phylogenetic ages (overall: 452.9 MY, immune response GO: 417.5 MY, stimulation induced: 397.9 MY, resampling $p < 1.5 \times 10^{-6}$, Fig. 1c). Similar trends were observed for alternative splice sites (overall: 292.5 MY, immune response GO: 247.7 MY, stimulation induced: 221.5 MY, resampling $p < 2.1 \times 10^{-4}$).

We subsequently searched for splice sites that differ between human and non-human primates, focusing on the 174,138 splice sites of coding exons that could be aligned in >80% of primate lineages. We identified 28 high-confidence human-specific splice sites (Supplementary Data 1), several of which involved immune genes such as the interferon-inducible gene *NUB1*, the leukocyte immunoglobulin-like receptor *LILRB4*, or the cytokine *CSF3* that is responsible for the regulation of granulocyte survival and proliferation (Fig. 1d). Altogether, these results support a more recent emergence of splice sites at immune genes, with respect to non-immune genes, and suggest that AS has been an important substrate of innovation in the evolution of human immune responses.

**Widespread impact of immune activation on isoform usage**. To characterise the variability of splice site usage in humans, we next quantified inter-individual differences in AS at the basal state and following immune stimulation. Focusing on highly expressed genes (FPKM > 10), we quantified 39,030 AS events and calculated the percent-spliced-in (PSI) index, which reflects the frequency at which a specific splicing event occurs. After quality checks and filtering of redundant AS events (Methods; Supplementary Fig. 2a), we obtained a final set of 16,173 independent AS events that affect a total of 4,739 genes (Fig. 2a; Supplementary Data 2, Supplementary Fig. 2b–d). Among AS events, 56% were associated with a switch between two functional protein-coding isoforms ($N = 9,098$, referred to as *modified protein*), 37% with a switch between a protein-coding and a non-coding isoform ($N = 6,067$, *gain/loss-of function*), and 6% with a switch between two non-coding isoforms ($N = 1,008$, *non-coding*). Among cases of gain/loss-of-function events, the non-coding isoform was lowly expressed in most cases (Supplementary Fig. 2e), suggesting strong constraints towards the expression of functional isoforms and/or a high rate of degradation of non-functional transcripts.

Principal component (PC) analysis, after adjusting PSI values for batch effects and technical variability (Methods; Supplementary Note 2; Supplementary Fig. 2f), revealed immune activation as the main source of splicing variability, with PC1 and PC2 reflecting the effects of IAV infection and TLR activation, respectively (Fig. 2b). Focusing on genes that change their

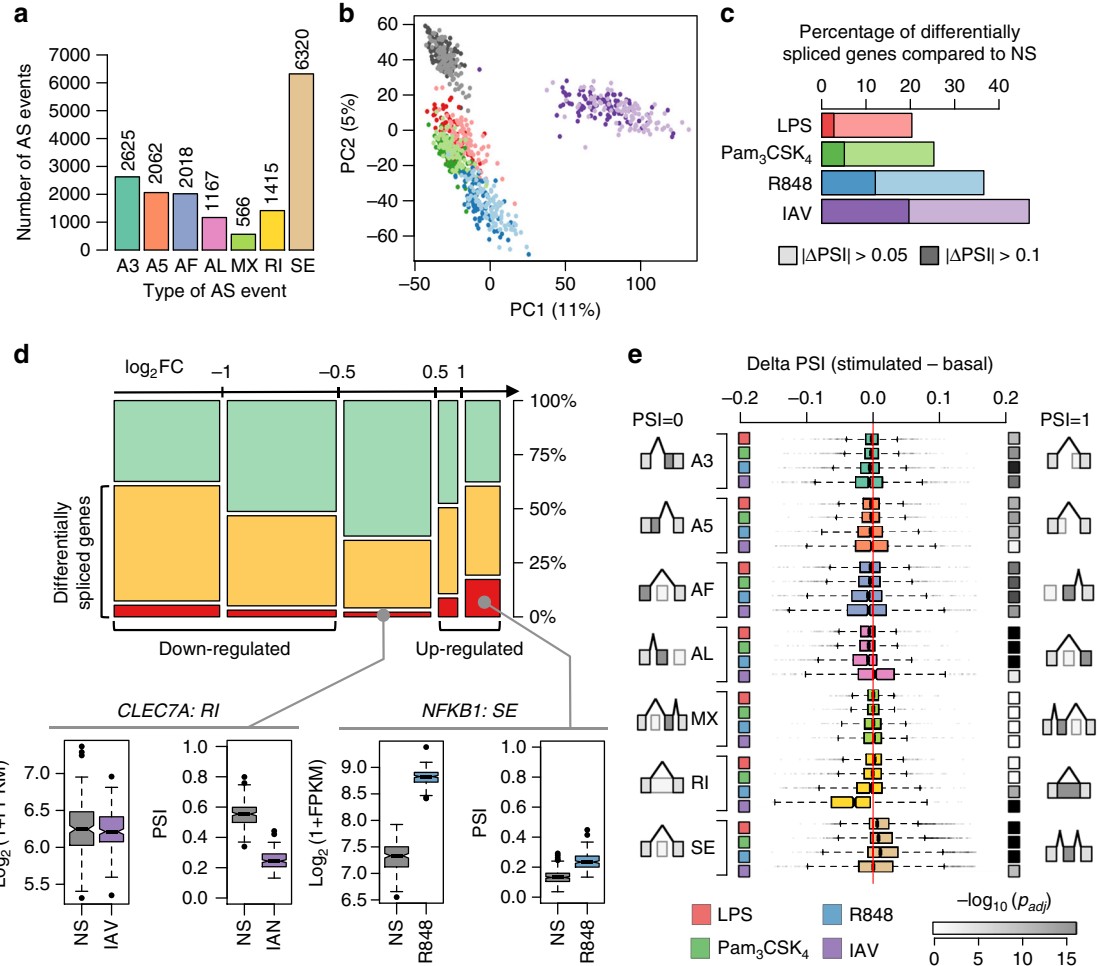

**Fig. 2** Immune stimulation induces major shifts in alternative splicing (AS) patterns. **a** Number of detected AS events by type: A3, alternative 3′ splice site; A5, alternative 5′ splice site; AF, alternative first exon; AL, alternative last exon; MX, mutually exclusive exons; RI, retained intron; SE, skipped exon. Each type of AS event is presented in a different colour that is consistently used across all figures. **b** Principal components analysis of AS events in different conditions: non-stimulated (grey), and activated with LPS (red), Pam$_3$CSK$_4$ (green), R848 (blue) and IAV (purple). Light and dark shades represent European and African individuals, respectively. The same colour code is used for conditions, across all figures. **c** Percentage of differentially spliced genes across conditions. **d** Percentage of genes differentially spliced after stimulation (in at least one condition), according to maximal fold-change in gene expression upon stimulation (log$_2$FC). Non-differentially spliced genes are shown in green, and differentially spliced genes in orange. Among the latter, immune response genes, as defined by GO, are highlighted in red. Levels of genes expression—log$_2$(1 + FPKM)—and alternative splicing—PSI—are shown for two immune genes, *CLEC7A* and *NFKB1*, at the basal state and in the stimulation condition where the changes in isoform usage are the most pronounced (centre line, median; box limits, upper and lower quartiles; whiskers, 1.5× interquartile range; points, outliers). Boxplots are coloured according to the condition of stimulation. **e** Boxplots showing the shifts in PSI values, for each type of AS event and stimulus (boxplots drawn as previously). The nature of the stimuli is indicated on the left-hand side by coloured squares and significance is indicated on the right-hand side by grey squares (darker for increased significance). Boxplots are coloured according to the type of AS event considered. Source data are provided as a Source Data file

isoform levels upon stimulation, we found 1,919 genes that were differentially spliced in at least one condition, with respect to the basal state (5% false discovery rate (FDR), |ΔPSI| > 0.05, Supplementary Data 2). A large number of differentially spliced genes (59%) displayed differential isoform usage in a stimuli-specific manner, with 789 being differentially spliced in a single condition (Supplementary Fig. 3a). The changes were markedly stronger following treatment with viral ligands (R848 and IAV), irrespectively of the |ΔPSI| threshold considered (Fisher's exact test for equal proportions; |ΔPSI| > 0.05; $p < 2.6 \times 10^{-23}$, |ΔPSI| > 0.1; $p < 6.2 \times 10^{-25}$, Fig. 2c).

We found that genes that are differentially expressed upon stimulation (i.e., |log$_2$FC| > 0.5) are more likely to change their isoform ratios (Fisher's exact test; OR = 2.2, $p < 4.34 \times 10^{-26}$, Fig. 2d). This involved key transcription factors controlling inflammatory and antiviral responses, such as *NFKB1* (change of

protein isoform upon all TLR stimulations, Wilcoxon $p < 2.5 \times 10^{-26}$, |ΔPSI| > 0.06, max |ΔPSI| = 0.11, log$_2$FC > 1.2, Fig. 2d) and *STAT2* (increased percentage of protein-coding transcripts upon R848 and IAV stimulations, Wilcoxon $p < 6.9 \times 10^{-41}$, |ΔPSI| > 0.1, max |ΔPSI| = 0.19, log$_2$FC > 0.8). Notably, we identified 460 genes presenting differential isoform usage after stimulation that were missed when searching for expression changes at the gene level (|log$_2$FC| < 0.2, Supplementary Data 2). An interesting example is provided by *CLEC7A*, a C-type lectin sensing β-glucans, which is differentially spliced across all stimulations (Wilcoxon $p < 2.9 \times 10^{-19}$, |ΔPSI| > 0.08), but does not change its expression in the IAV stimulation where the changes in splicing are the strongest (increased percentage of protein-coding transcripts, max |ΔPSI| = 0.28, Fig. 2d). These results reveal a wide array of AS modifications upon monocyte exposure to immune challenges, and highlight the importance

of considering isoform modifications in the context of immune function.

**Understanding the nature of isoform changes upon stimulation.** We next explored the qualitative nature of isoform changes in response to immune or infectious challenges. We found systematic shifts in splicing patterns upon stimulation (Fig. 2e), including increased inclusion of cassette exons (skipped exon class, SE, Wilcoxon $p < 2.9 \times 10^{-124}$) and usage of upstream transcript termination sites (alternative last exon class, AL, Wilcoxon $p < 5.4 \times 10^{-23}$) as observed in response to bacterial infections[18]. In addition, we detected a decrease in intron retention (retained intron class, RI, Wilcoxon $p < 5.9 \times 10^{-67}$) that was specific to IAV stimulation. Consistent with previous findings[17–20], these qualitative changes were associated to an increased usage of the minor isoforms upon stimulation, leading to higher isoform diversity upon immune challenge for 70% of the genes.

Focusing on gain/loss-of-function AS events, we observed a strong shift towards an increase in non-coding isoforms upon all stimulations (Wilcoxon $p < 1.1 \times 10^{-29}$, Supplementary Fig. 3b). However, such an increase was found to be less common among upregulated genes (Fisher's exact test; $OR = 0.68$, $p < 0.008$) and more frequent among downregulated genes (Fisher's exact test; $OR = 1.8$, $p < 1.2 \times 10^{-9}$) (Supplementary Fig. 3c). This observation suggests a contribution of AS-induced nonsense-mediated decay (AS-NMD) to the regulation of gene expression. To assess

the extent to which AS-NMD may play an active role to downregulate gene expression or simply result from splicing errors, we focused on exons whose inclusion leads to a nonsense isoform. We reasoned that splice sites that are used to actively regulate gene expression through NMD are likely to be evolutionary conserved, whereas those that disrupt essential proteins will tend to be removed by natural selection. We found that 35% of the exons examined have at least one conserved splice site, consistent with an active role in downregulating gene expression (Supplementary Fig. 3d). Together, these findings increase our understanding of the nature of isoform changes in response to immune stimuli, and support a role of AS as an additional layer of immune response regulation through AS-NMD.

**Increased noisy splicing upon immune activation.** The increase in non-coding isoforms detected in response to immune stimulation could also reflect noisy splicing (i.e., splicing leading to non-functional transcripts)[22–24]. To explore this possibility, we quantified the frequency at which constitutive donor or acceptor sites are joined with cryptic (unannotated) splice sites, as a measure of mis-splicing events. We then defined the rate of noisy splicing of a gene as the average of these frequencies across all its constitutive splice sites (Fig. 3a). At the basal state, ~0.29% of splicing events on average across genes corresponded to noisy splicing (Fig. 3b; Supplementary Data 2), a figure that ranged from <0.001% (6% of multi-exonic genes) to >10% (0.2% of multi-exonic genes). Indeed,

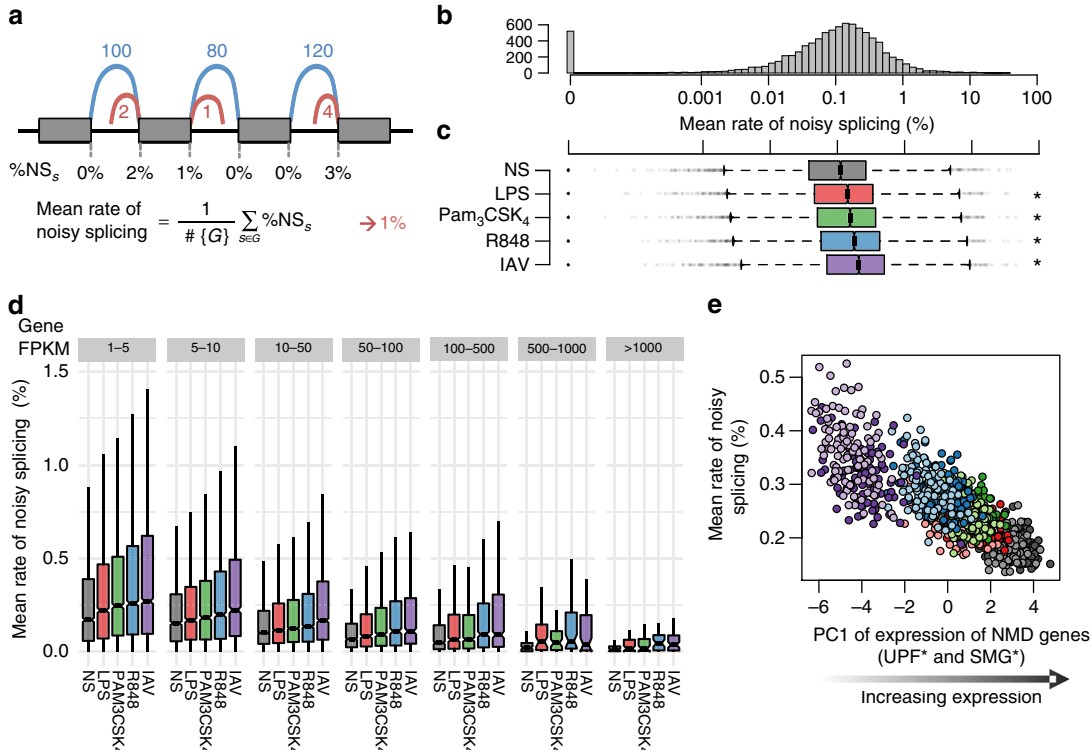

**Fig. 3** Increase in noisy splicing upon immune stimulation. **a** For each constitutive splice site neighbouring a coding exon, functional splicing events (blue) that join two exons are distinguished from non-functional splicing events (red) where the splice site is joined with a cryptic, non-conserved splice site (GerpRS < 2). For each gene, the mean rate of noisy splicing was computed as the average across all constitutive splice sites of the frequency at which non-functional splicing events occur. **b** Distribution of the rate of noisy splicing at the basal state. **c** Distribution of the levels of noisy splicing per gene according to the condition of stimulation (centre line, median; box limits, upper and lower quartiles; whiskers, 1.5× interquartile range; points, outliers). The significance of differences between noisy splicing in each stimulation condition and the basal state was assessed by a Wilcoxon rank test (*$p < 10^{-20}$). **d** Distribution of the rate of noisy splicing per gene as a function of gene expression. For each bin of expression, the rate of noisy splicing is shown across all conditions of stimulation (boxplots drawn as previously). **e** Average rate of noisy splicing per sample correlates with expression of nonsense-mediated decay genes (first PC). For each sample, the colour reflects the condition of stimulation (grey: NS, red: LPS, green: Pam3CSK4, blue: R848, purple: IAV). Light and dark shades indicate European and African individuals respectively. Source data are provided as a Source Data file

noisy splicing increased with mean intron length (Spearman's $\rho = 0.33$, Student's $p < 2.8 \times 10^{-231}$), and decreased with gene expression levels (Spearman's $\rho = -0.23$, Student's $p < 4.6 \times 10^{-109}$). Interestingly, noisy splicing also decreased with levels of nascent mRNAs (measured from intronic reads, Spearman's $\rho = -0.19$, Student's $p < 1.8 \times 10^{-74}$), suggesting coupling between transcription rate and splicing efficiency. However, gene-to-gene differences in transcription rate could not fully account for the correlation between noisy splicing and gene expression levels (Spearman's $\rho = -0.16$, Student's $p < 3.2 \times 10^{-50}$ after adjusting on mean intronic coverage), consistent with the degradation by NMD of non-functional transcripts produced through noisy splicing.

Upon stimulation, we found that noisy splicing increased by 15–67% on average (Wilcoxon $p < 5.1 \times 10^{-22}$, Fig. 3c), with this increase being detected across all ranges of gene expression (Wilcoxon $p < 5.1 \times 10^{-5}$, Fig. 3d). To verify that this observation was not explained by the improved detection of cryptic splice sites after stimulation due to higher gene expression, we measured the change in the number of cryptic splice sites and noisy splicing per gene, according to fold changes in gene expression upon stimulation. We found that while the detection of cryptic splice sites does correlate with gene expression, the rate of noisy splicing increases for both up and downregulated genes (Supplementary Fig. 4). We further observed that the increase in noisy splicing detected upon stimulation was associated to a decreased expression of NMD genes (Spearman's $\rho = -0.81$, Student's $p < 4.5 \times 10^{-226}$, Fig. 3e). These results indicate that while splicing errors increase in response to immune stimulation due to reduced NMD activity, such errors are less common amongst highly transcribed genes, thereby mitigating their impact on immune gene expression.

**Characterising the genetic bases of splicing regulation.** To investigate the genetic regulation of splicing variability, we mapped sQTLs, i.e., genetic variants associated with changes in AS. We tested the 16,173 AS events for association, in *cis*, with 5,634,819 single nucleotide polymorphisms (SNPs) at a minor allele frequency (MAF) > 0.05, located within a 1 Mb-window of AS events. At a 5% FDR, we identified 1,464 AS events significantly associated with a SNP, corresponding to 21% of the 4,739 AS genes (Supplementary Data 3). We found an excess of sQTLs among immune response genes (non-central hypergeometric test (GOseq); OR = 1.7, $p < 1.5 \times 10^{-8}$), receptors (OR = 2.3, GOseq $p < 3.2 \times 10^{-6}$) and genes located at the cell periphery (OR = 1.5, GOseq $p < 2.5 \times 10^{-9}$) (Supplementary Data 3). These enrichments remained significant when excluding the *HLA* region (GOseq $p < 2.4 \times 10^{-5}$), and accounting for gene expression levels (resampling $p < 1.0 \times 10^{-4}$) or a higher rate of AS events at immune response genes (resampling $p < 1.0 \times 10^{-4}$).

After excluding the *HLA* region, 66% of sQTLs were located within 10 kb of their respective AS event (Fig. 4a), with 38% falling directly within the boundaries of the AS event. We found that sQTLs were strongly enriched in exonic and near-exonic regions (<300 nucleotides from splice site, OR > 5.7, Fisher's exact $p < 1.5 \times 10^{-84}$), and in loci predicted to alter splicing (SPIDEX database[36], OR = 4.7, Fisher's exact $p < 1.7 \times 10^{-19}$, Fig. 4b). Consistently, we found a strong enrichment of sQTLs in splicing regulatory regions (e.g., branchpoint, OR = 17.9, Fisher's exact $p < 3.5 \times 10^{-75}$), RNA-binding motifs (e.g., RBM4; OR = 4.5, Fisher's exact $p < 3.1 \times 10^{-96}$), highly conserved sites (GerpRS > 4; OR = 3.1, Fisher's exact $p < 1.7 \times 10^{-7}$), as well as promoter regions (OR = 8.6, Fisher's exact $p < 4.61 \times 10^{-94}$; Supplementary Fig. 5a).

**Untangling the genetic control of splicing and transcription.** We next investigated whether the same regulatory variants were associated with changes in splicing (sQTLs) and gene expression (expression quantitative trait loci (eQTLs)) (Methods; Supplementary Data 3). Remarkably, 11% of sQTLs overlap an eQTL ($N = 155$, $r^2 > 0.8$, Fig. 4c), and ~35% of sQTLs ($N = 513$) showed marginal associations with the expression levels of the gene they regulate ($p_{eQTL} < 0.05$). Three scenarios can explain the link between splicing and gene expression: (i) changes in isoform levels may alter gene expression through NMD, (ii) transcription could lead to AS by activating, for example, an alternative transcription start site (TSS) or by increasing transcriptional elongation rate and preventing exon recognition, and (iii) the co-occurrence of a sQTL with an eQTL results from linkage disequilibrium (LD) between variants that regulate splicing and transcription[1,20,43].

To discern between these scenarios, we used a two-step approach; we first tested for a link between splicing and gene expression when accounting for the effects of genetics, to subsequently assess the direction of this link by comparing the likelihood of each causality model[44] (Methods; Supplementary Fig. 6; Supplementary Note 3). Among the 155 sQTLs overlapping an eQTL, splicing remained significantly correlated to gene expression, after adjusting for genotype, in ~70% of the cases (Fig. 4d). Among these, we identified 21 sQTLs where splicing modifications lead to gene expression changes (causal sQTLs, ~14% of tested sQTLs). For example, the rs2927608 variant at *ERAP2* is associated with an increased proportion of nonsense isoforms (Student's test; $\beta_{sQTL} = -0.3$, $p_{sQTL} = 1.7 \times 10^{-66}$) that lead to decreased gene expression via NMD (Student's test; $\beta_{eQTL} = -0.6$, $p_{eQTL} = 3.2 \times 10^{-53}$)[19].

However, the most frequent scenario (~57% of tested sQTLs) was that of a reactive sQTL (Supplementary Fig. 6), where splicing changes are mediated by gene expression. This is illustrated by the rs1317397 variant, located in the promoter flanking region of the chemokine receptor *CMKLR1*, which is associated to a strong decrease in expression (Student's test; $\beta_{eQTL} = -0.6$, $p_{eQTL} = 1.6 \times 10^{-20}$) that leads to an increased inclusion of exon 3 (Student's test; modified protein, $\beta_{sQTL} = 0.03$, $p_{sQTL} = 7.3 \times 10^{-10}$). Extending our analysis to the 513 sQTLs that present marginal associations with gene expression but do not overlap an eQTL, we found that the occurrence of causal sQTLs increased to 34%, whereas that of reactive sQTLs decreased to ~5% (Fig. 4d). This suggests that the majority of causal sQTLs have a moderate effect on gene expression.

Finally, to assess whether reactive sQTLs could act through the modulation of transcriptional elongation rates, we analysed the effect of sQTLs on transcription rate, using intronic reads as a proxy for nascent mRNA levels[45]. We detected a significant effect for 89% of reactive sQTLs, compared with 25% of sQTLs that do not correlate with expression (OR = 24.2, Fisher's exact $p < 9.1 \times 10^{-28}$). Together, these analyses indicate that the genetic determinants of isoform usage and gene expression are largely independent; yet, when they are not, the regulation of AS through modulation of the transcription rate seems to be the predominant model.

**Mapping the genetic bases of context-specific splicing.** We subsequently explored the context specificity of sQTLs, using a Bayesian model selection approach (Supplementary Note 4) to identify cases where the genetic regulation of isoform usage is altered depending on the presence or absence of immune stimuli. Focusing on genes expressed in all conditions, we found that 71% of sQTLs ($N = 684$) were shared across all conditions, whereas 5.3% of them ($N = 51$) were detected in a single condition (Fig. 4e; Supplementary Fig. 5b).

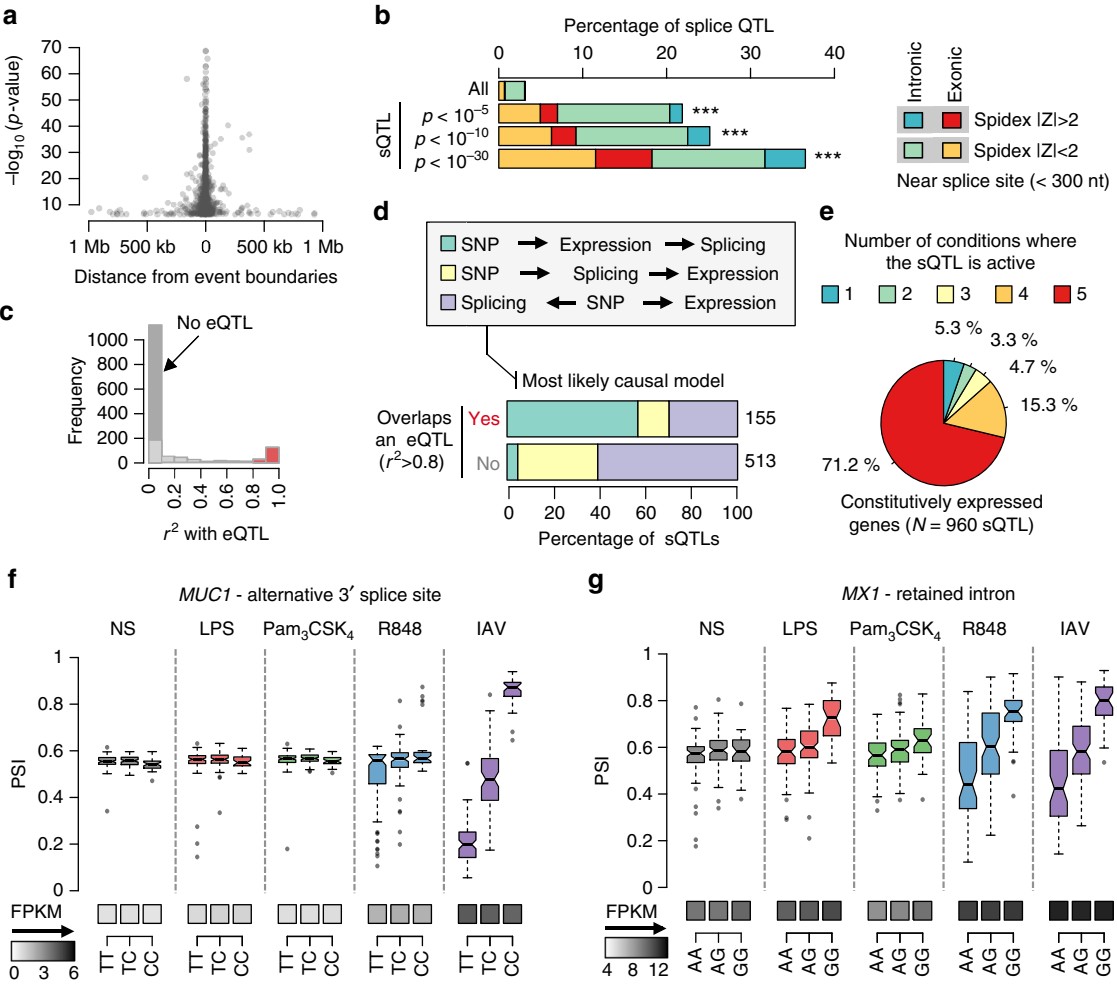

**Fig. 4** Genetic regulation of splicing variation in response to immune stimulation. **a** Distance of splice QTLs (sQTLs) to their associated AS event. Most sQTLs are located in close proximity to their respective AS event. **b** Enrichment of sQTLs in near-exonic regions and sites predicted to alter splicing by SPIDEX. Twenty percent of sQTLs are located in exonic or intronic regions located <300 nucleotides (nt) apart of a splice site. The enrichment increases with the levels of significance of sQTLs (***Fisher's $p < 1.1 \times 10^{-16}$). **c** Overlap between all sQTLs and eQTLs identified for the same gene and condition. sQTLs of genes without eQTLs are shown in dark grey, and sQTLs that colocalize with an eQTL ($r^2 > 0.8$) are shown in red. **d** Frequency of inferred causal model among sQTLs associated with gene expression. Causality is shown separately for sQTLs overlapping a *cis*-eQTL ($r^2 > 0.8$), and sQTLs that show marginal associations with gene expression ($p_{expr} < 0.05$ in the condition where the sQTL is strongest) but do not overlap an eQTL. For each group, the total number of sQTLs is reported. **e** Sharing of sQTLs across conditions. For the 960 sQTLs where the corresponding gene is expressed across all conditions, the pie chart represents the number of conditions where the sQTL is active. **f** Example of a context-specific sQTL that manifests only upon IAV infection because the expression of the corresponding *MUC1* gene is restricted to the IAV condition. For each genotype and condition, the boxplots show the distribution of PSI (centre line, median; box limits, upper and lower quartiles; whiskers, 1.5× interquartile range; points, outliers). For each genotype and conditions, grey squares indicate FPKM levels (darker shade = higher gene expression). **g** Response sQTL (rsQTL) leading to differential intron retention in the 5′-UTR region of the *MX1* gene in response to LPS, R848 and IAV (boxplots drawn as previously). **f**, **g** Boxplots are coloured according to the condition of stimulation. Source data are provided as a Source Data file

We then searched for sQTLs that manifest only in the presence of immune stimuli (Supplementary Data 3), because either the gene is expressed only upon immune challenge (sQTLs of stimulation-specific genes) or the genetic variant alters splicing in a stimulation-specific manner (response sQTLs, rsQTLs) reflecting G × E interactions. Focusing of the 274 stimulation-specific genes (FPKM < 10 at basal state and $\log_2$FC > 1 in at least one condition), we detected 74 genes with at least one sQTL, for a total of 108 sQTLs. For example, the rs4072037 variant induces a change in 3′ splice sites of *MUC1*, encoding a protein at the cell surface of lung epithelial cells, which is expressed only upon IAV infection (modified protein, Student's test; $p_{sQTL} < 1.1 \times 10^{-54}$, Fig. 4f). To search for rsQTLs, we focused on genes expressed both at the basal state and after stimulation (FPKM > 10), and found

127 rsQTLs showing a significantly stronger effect in response to stimulation ($p_{rsQTL} < 0.01$, corresponding to ~5% FDR). For instance, the influenza resistance gene *MX1*, while being expressed at basal state, displays genotype-dependent intron retention specifically upon stimulation by LPS, R848 or IAV (rs462687, modified protein, Student's test; $p_{rsQTL} < 6.7 \times 10^{-16}$, Fig. 4g). These examples highlight how cellular perturbation with external stimuli can reveal, or alter, the genetic regulation of AS in the context of immunity to infection.

**AS contributes to immune disorders**. To understand how splice regulatory variants may ultimately impact organismal traits, we searched for overlaps between sQTLs and loci associated with complex traits or diseases by genome-wide association studies

(GWAS)[46]. We identified 195 sQTLs that overlap GWAS hits (Methods; Supplementary Data 3). When grouping traits according to their Experimental Factor Ontology (EFO) category[47], we found that loci associated with immune system disorders such as ulcerative colitis or type-1 diabetes ($N = 27$, OR = 4.6, resampling $p < 0.001$) and body measurements such as height or body mass index ($N = 25$, OR = 3.1, resampling $p < 0.001$) were the most significantly enriched (Fig. 5a).

Among disease-associated sQTLs at the basal state, we identified a variant (rs2271543) that promotes the inclusion of a 5′UTR exon in *TMBIM1* (modified protein, Student's test; $p_{sQTL} < 3.1 \times 10^{-33}$, $R^2 > 0.54$, |SPIDEX's $Z$| = 2.1) and colocalizes

with a susceptibility locus for inflammatory bowel disease[48] (Fig. 5b). Similarly, a variant (rs1333973) affecting the exclusion of exon 2 of the antiviral gene *IFI44L* (modified protein, Student's test; $p_{sQTL} < 1.1 \times 10^{-56}$, $R^2 = 0.72$, SPIDEX's $Z = 3.9$) is associated with measles-specific humoral immunity and increased risk of febrile seizures[49,50]. Finally, a non-synonymous variant (rs1127354) leading to the removal of the second exon of *ITPA* (modified protein, Student's test; $p_{sQTL} < 2.8 \times 10^{-12}$, $R^2 > 0.22$, |SPIDEX's $Z$| = 2.6) associates with increased susceptibility to hepatitis C virus infection and ribavirin-induced anaemia[51].

Focusing on the 235 sQTLs that manifest only upon immune stimulation, we found 20 sQTLs of stimulation-specific genes

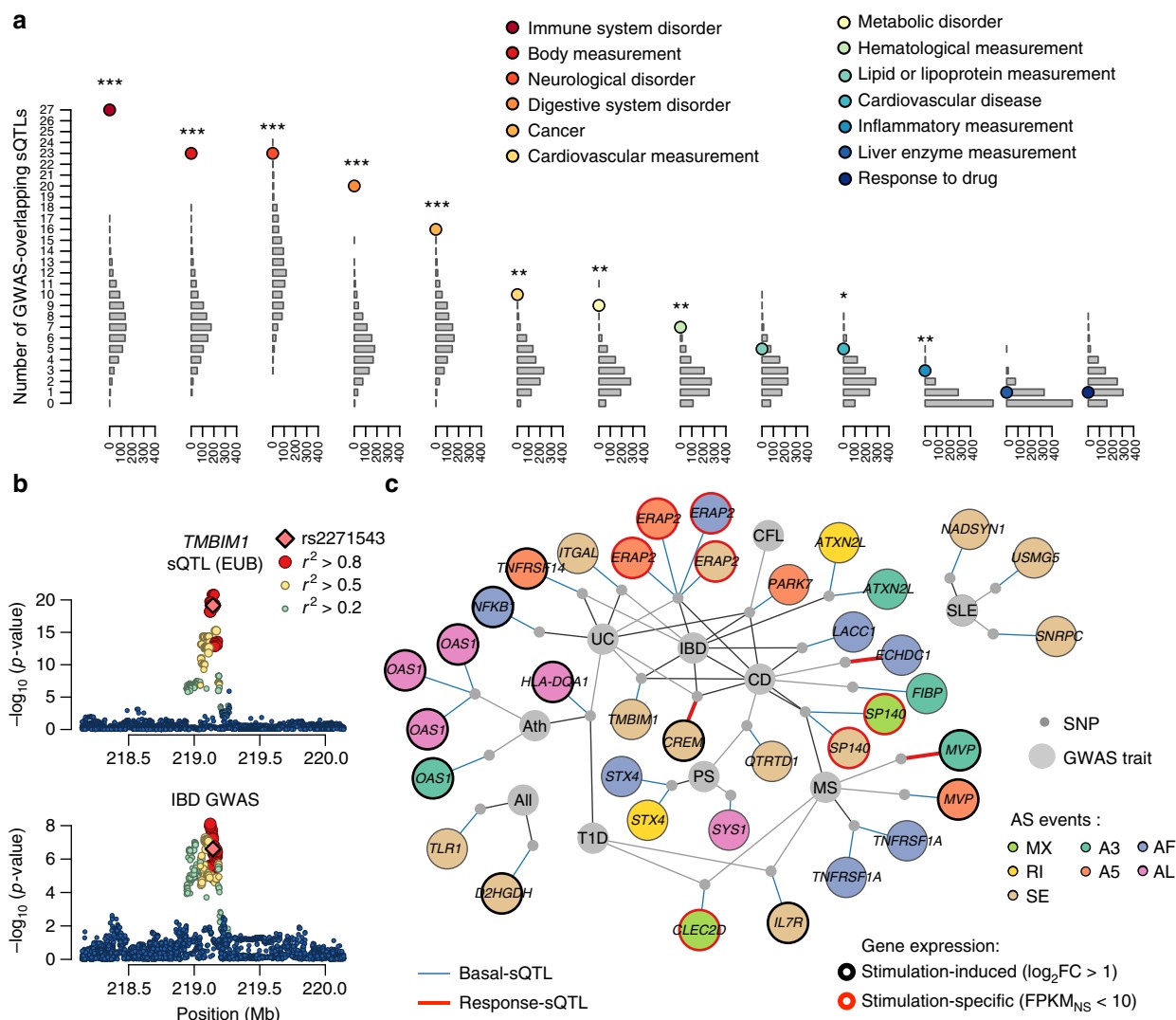

**Fig. 5** Innate immunity sQTLs contribute to complex traits and immune disorders. **a** Enrichment of sQTLs in GWAS hits. For each trait/disease category, coloured dots show the number of sQTLs overlapping a GWAS locus. Grey bars represent the distribution of the expected number of GWAS hits when resampling 1000 random SNPs matched for minor allele frequency (resampling $p$-value, *$p < 0.05$, **$p < 0.01$, ***$p < 0.001$). **b** Log$_{10}$($p$-value) profiles of the *TMBIM1* sQTL (upper panel) and GWAS for inflammatory bowel disease (IBD, lower panel). SNPs are coloured according to their LD ($r^2$) with the sQTL peak SNP and putative causal variant rs2271543. Only SNPs where both sQTLs and GWAS summary statistics were available are shown. **c** sQTLs colocalizing with GWAS loci for 10 major immune-related disorders (grey nodes, Ath asthma, All allergy, CEL celiac disease, CD Crohn's disease, IBD inflammatory bowel disease, MS multiple sclerosis, PS psoriasis, SLE systemic lupus erythematous, T1D type-1 diabetes, UC ulcerative colitis). For each sQTL, genetic variants are represented by grey dots and are linked to the associated AS event (coloured nodes, named after their associated gene). AS events are coloured by class of event and circled according to gene expression patterns upon stimulation (black: upregulated—log$_2$FC > 1 in at least one condition, red: specific to stimulation—FPKM < 10 at basal state and log$_2$FC > 1). Links between AS events and sQTLs indicate stimulation-specificity of the genetic association (blue edge for sQTLs, red for response sQTLs—$p$-value > 0.001 at basal state and $p_{rsQTL} < 0.01$). Links between sQTLs and GWAS traits indicate colocalization ($r^2 > 0.8$ between sQTLs and GWAS SNPs in Europeans) and are coloured according to the strength of the GWAS association (grey: GWAS $p < 10^{-5}$, black: GWAS $p < 10^{-8}$). Source data are provided as a Source Data file

and 14 rsQTLs that overlap GWAS hits. Of these 34 sQTLs, 12 were associated with immune disorders (Fig. 5c; Supplementary Data 3). In addition to known disease-causing sQTLs at *SP140* or *ERAP2*[19,52,53], we detected novel cases at immune genes such as *MVP* and *CLEC2D*, which encode key regulators of the JAK/STAT and interferon-γ pathways, respectively. The *MVP* variant (rs13332078) promotes the usage of an alternative 3′ splice site upon stimulation with R848 (modified protein, Student's test $p_{rsQTL} < 0.007$), and is associated with increased risk of multiple sclerosis[54]. The *CLEC2D* variant (rs7968401), involving a switch between mutually exclusive exons after TLR activation (modified protein, Student's test $p_{sQTL} < 9.4 \times 10^{-10}$, $R^2 = 0.17$), overlaps hits associated with multiple sclerosis and type-1 diabetes[55,56]. These results not only support the importance of AS as a mechanism explaining variation of complex traits[2,19,20,27,35,36], but also highlight immune pathways where differential isoform usage following cellular perturbation may influence immune disease outcomes.

**Differential splicing as a substrate of local adaptation.** To investigate the role of splicing as a mechanism of local adaptation, we first searched for AS events that presented different PSI values between African- and European-descent individuals (FDR < 5%, |ΔPSI| > 0.05). We found 515 genes that were differentially spliced between populations in at least one condition (pop-DSG, Supplementary Data 4), of which 60% ($N = 309$) were detected only after stimulation (Supplementary Fig. 7a). Among pop-DSG presenting the largest population differences (|ΔPSI| > 0.1), we found key regulators of the immune response, such as the microbial sensor *NOD1*, upon R848 and IAV stimulation (modified protein, Wilcoxon $p < 6.5 \times 10^{-20}$, |ΔPSI| > 0.27, max |ΔPSI| = 0.37), and the transcription factor *NFKB1*, after all TLR stimulations (modified protein, Wilcoxon $p < 6.1 \times 10^{-12}$, |ΔPSI| > 0.16, max |ΔPSI| = 0.22). Furthermore, pop-DSG were strongly enriched in sQTLs, both at the basal state and after stimulation (OR > 8.1, Fisher's $p < 3.5 \times 10^{-88}$, Fig. 6a). Using causal mediation analysis[57], we found that among pop-DSG with sQTLs, differences in sQTL allele frequency between populations account for 67% of population differences in splicing levels, a figure that reached 78% for the strongest sQTLs (i.e., $\beta_{sQTL} > 0.2$, Fig. 6b and Supplementary Fig. 7b).

Once established that a large fraction of population differences in splicing can be attributed to genetic factors, we searched for sQTLs that presented signatures of positive selection, based on levels of population differentiation ($F_{ST}$) and extended haplotype homozygosity (integrated haplotype score (iHS))[58,59]. To reduce false positives, we considered as selection candidates only loci presenting both the strongest selection signatures (>95th percentile genome-wide for $F_{ST}$ or |iHS|) and an enrichment of SNPs with such signals around the sQTL[59,60]. In doing so, we detected 29 sQTLs presenting extreme population differentiation ($F_{ST} > 0.4$, Supplementary Data 4). Among these hits, the strongest $F_{ST}$ was found at the RAB GTPase activator *RABGAP1* (rs59393793, $F_{ST} = 0.84$, $p_{emp} = 1.2 \times 10^{-4}$, beta-binomial test $p_{enrich} < 0.01$), with Europeans showing an increased frequency of the derived allele associated with a diminished amount of the protein-coding isoform. Focusing on immune genes, the sQTL with the strongest $F_{ST}$ was found for the *CD226* locus, encoding a glycoprotein involved in T-cell activation (rs1823778, $F_{ST} = 0.75$, $p_{emp} < 9.2 \times 10^{-4}$, beta-binomial test $p_{enrich} < 0.007$), also showing a higher frequency of the derived allele in Europeans but, in this case, leading to a qualitative change in the resulting protein-coding isoform.

With respect to |iHS| signals, which reflect more recent events of positive selection[59], we detected 33 sQTLs presenting outlier

values (13 in European and 20 in Africans, Supplementary Data 4). Among these, the strongest signals of adaptation were found for the lysozyme regulator *HEATR7A* in Africans (rs6558326, |iHS| = 3.1, $p_{emp} = 0.003$, beta-binomial test $p_{enrich} < 0.05$), and for the meiosis regulator *CPEB2* in Europeans (rs12502866, |iHS| = 3.6, $p_{emp} = 0.0007$, beta-binomial test $p_{enrich} < 0.003$). The largest |iHS| among immune genes was found at the *P2RX7* locus (rs208293, |iHS| = 2.95 in Europeans, $p_{emp} = 0.003$, beta-binomial test $p_{enrich} < 0.05$), which is involved in inflammasome activation and intracellular pathogen clearance[61].

Among sQTLs of stimulation-specific genes, we found the strongest signal of selection for the rs776746 variant, which C allele promotes the inclusion of an additional exon in *CYP3A5* upon all TLR stimulations. This allele, which decreases the percentage of protein-coding isoform, presents a very high frequency in Europeans ($F_{ST} = 0.73$, $p_{emp} = 0.001$, beta-binomial test $p_{enrich} < 0.005$, Fig. 6c–e; Supplementary Fig. 7c). Interestingly, the C allele has been reported to protect against paediatric tuberculosis[62] and, using the UK Biobank atlas of genetic associations[63], we found it associated with varying proportions of neutrophils, reticulocytes and lymphocytes, and increased risk of allergic rhinitis (Fig. 6f). Together, these results highlight the important role of splicing not only as a source of population variation of immune responses but also as a potential driver of local adaptation.

**Archaic introgression contributed to splicing variability.** Because admixture between modern humans and Neanderthals has been shown to contribute to present-day population differences in immune responses[29,38,64], we explored how Neanderthal introgression has affected splicing. We overlapped our list of sQTLs with a set of 100,755 SNPs putatively brought into European genomes by Neanderthal introgression (Supplementary Note 5). We detected eight loci where the presence of Neanderthal haplotypes promotes differential isoform usage (Supplementary Data 4), representing a 2.8-fold enrichment with respect to genome-wide expectations (resampling $p < 7.0 \times 10^{-3}$).

Among archaic sQTLs, we observed the previously described rs10774671 associated with AS at *OAS1* (ref. [53]). We also found novel cases, including sQTLs at the microbial sensor *TLR1* (rs5743593, archaic allele associated with a change in protein-coding isoform, Student's test $p < 2.1 \times 10^{-11}$ in the non-stimulated state, $MAF_{EU} = 0.165$), the cytosolic sensor *DDX60L* (rs56999040, decreasing protein-coding transcripts, $p < 3.2 \times 10^{-8}$ in the non-stimulated state, $MAF_{EU} = 0.165$), shown to inhibit viral replication in vitro[65], and the Fc-γ receptor *FCGR2A* (rs373579, increasing protein-coding transcripts, $p < 1.6 \times 10^{-9}$ after stimulation by LPS, MAF = 0.14), triggering phagocytosis of bacterial antigens after IgG stimulation[66]. Overall, these results show that introgression of splice regulatory variants from Neanderthals has also contributed to diversify the patterns of immune response variation currently observed among individuals of European ancestry.

**Discussion**
Several important insights can be drawn from our study. First, we show that activation of major innate immunity pathways, such as TLR1/2, TLR4 and TLR7/8, and infection with a live strain of IAV increase isoform diversity and elicit a marked remodelling of the isoform repertoire. Our results are consistent with previous findings of global shifts in response to bacterial and viral stimuli detected in macrophages or dendritic cells[17–20], but show that such an elevated isoform diversity can be attributed, to a large extent, to a global increase in splicing

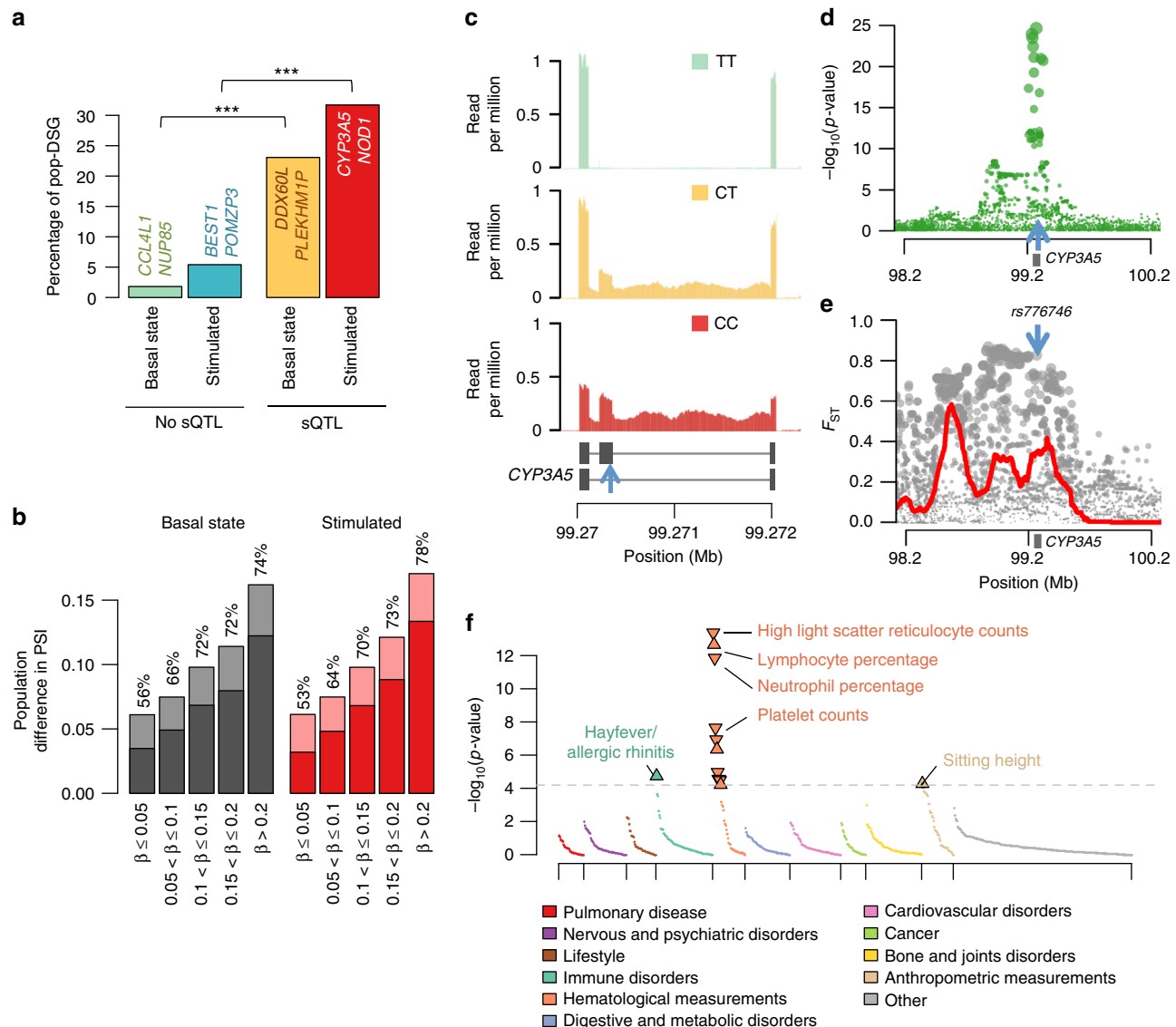

**Fig. 6** Population differences in alternative splicing and impact of positive selection. **a** Percentage of genes showing differential splicing between populations (pop-DSG) at the basal state and after stimulation, classified according to the presence of a sQTL (Fisher's exact test for equal proportion, ***$p < 10^{-20}$). For each group, examples of genes that present the strongest population differences are reported. For the stimulated state, the reported genes are chosen among those that are differentially spliced specifically after stimulation. **b** Contribution of genetic factors to the detected population differences in splicing patterns. The mean difference in PSI value between populations is represented, at the basal state (grey) and after stimulation (red), for genes presenting an sQTL with various levels of effect size (β). For each group, the average difference in PSI value that is mediated by the sQTL is shown in dark shade, and the corresponding percentage is indicated above the bar. **c** The derived allele (C) of the sQTL rs776746 (blue arrow) is associated with the inclusion of an additional exon at the *CYP3A5* gene. **d** Association between SNPs in a 2 Mb window around the sQTL rs776746 and splicing levels of *CYP3A5* in the Pam₃CSK₄ condition. **e** Signals of positive selection around the *CYP3A5* locus, measured by the $F_{ST}$ metric. The red curve shows the proportion of SNPs with extreme $F_{ST}$ values (>95% percentile), in a 100 kb window around each SNP. **f** Phenome-wide association study of the SNP rs776746, based on 778 phenotypic traits measured by the UK biobank study[63]. The dashed line indicates the Bonferroni-corrected significance threshold. For significantly associated traits, arrows indicate the direction of the effect of the derived allele (**c**). Source data are provided as a Source Data file

errors leading to non-functional transcripts. Erroneous splicing has been previously detected in lymphoblastoid cell lines, in the absence of any stimulation[23,24]. We find that noisy splicing increases not only upon stimulation by live pathogens but also with synthetic stimuli, such as TLR ligands, suggesting that this phenomenon is driven by the host itself. A plausible explanation could be a reduction in the activity of NMD upon stimulation, leading to reduced degradation of mis-spliced transcripts. This is consistent with the increase in the number of

truncated/non-translatable transcripts reported after infection of macrophages by *Mycobacterium tuberculosis*[17]. It has been shown that highly expressed genes are less prone to erroneous splicing, following the removal of genetic variants leading to non-functional isoforms by purifying selection[24]. Our findings indicate that, although immune stimulation increases the degree of splicing errors, highly expressed genes tend to maintain a reduced error rate after stimulation, allowing for an efficient immune response.

We also provide evidence for condition-specific genetic regulation of isoform usage. We found a total of 993 genes with evidence of genetic control of AS in resting or stimulated monocytes. Although such genetic regulation was largely shared across conditions, we detected a large number of regulatory variants acting in a context-dependent manner (up to 28% of the total number of detected sQTLs). Notably, ~9% of sQTLs altered splicing only in the presence of immune stimulation, regardless of gene expression levels, suggesting the occurrence of G × E interactions. Furthermore, our analyses indicate that the genetic control of AS is largely independent to that of transcription, as hitherto observed[19,20,25,27,31], but extend previous findings by inferring causality between regulatory mechanisms when these are not independent. Through the analysis of the conditional independence structure between genotypes, gene expression and isoform ratios, we show that changes in splicing are primarily mediated by changes in gene expression, rather than the opposite. This finding, supported by our analysis of nascent transcript abundance, illustrates the complex interactions between the transcription and splicing machineries[1] and highlights the contribution of variants that modulate transcription to AS.

From an evolutionary perspective, this study shows decreased inter-species conservation of immune gene splicing, which is due to an increased rate of splice site emergence at immune genes, as attested by the human-specific splice sites detected at *CSF3* and *NUB1*. Further support to the lower constraint acting on immune gene splicing comes from their enrichment in sQTLs, with respect to the remainder of the coding genome. Given that the immune system is our primary interface with the environment, including pathogens[8–10], these findings may also support a selective model that favours diversity. Under such a model, while most variants leading to non-functional transcripts (i.e., noisy splicing) are purged by purifying selection[24], genetic variants that alter splicing of immune genes may also be favoured by different forms of adaptive evolution, allowing for a broader repertoire of functional isoforms.

The signals of positive selection we detect at splicing regulatory variants and their enrichment in Neanderthal ancestry further support that differential isoform usage can be a vehicle for local adaptation, or at least contribute to diversify immune responses. For example, the strong frequency increase of the rs776746-C allele at the *CYP3A5* locus in Europeans, which triggers the inclusion of an additional exon in response to TLR activation, may reflect the advantageous nature of this variant. Plausible adaptive phenotypes associated with the rs776746-C allele include xenobiotic-metabolism, sodium-sensitive hypertension or decreased susceptibility to paediatric tuberculosis[62,67]. Similarly, our analyses reveal that the high Neanderthal ancestry we and others have detected at the *TLR1* locus[68,69] is related to the presence of various alleles that alter *TLR1* isoform usage. Together, these results suggest a role of AS as a substrate of human adaptation, following various evolutionary mechanisms that operate at different time-scales. Additional simulation-based work, accompanied by studies of ancient DNA time transects for the direct detection of natural selection (i.e., analyses of temporal changes of allelic frequencies), is now required to validate the action of positive selection at the sQTLs identified.

The case of *CYP3A5* also illustrates how past adaptation can have deleterious consequences in the present day, as the putatively selected rs776746-C allele is associated with increased allergic sensitivity in Europeans[63]. The strong enrichments of sQTLs in variants associated with complex diseases[19,20,27,35], in particular chronic inflammatory disorders in our setting, suggest the contribution of splice regulatory variants to the genetic architecture of deleterious phenotypes. Future work using colocalization analyses and Mendelian randomisation approaches should help establishing a causal role of AS in disease risk at the loci identified. As such, the regulatory variants we report here, whether adaptive or not, constitute a valuable resource to explore further how AS contributes to individual and population variability in host responses to infection and immune disease susceptibility.

## Methods

**Ethics statement.** All experiments involving human primary monocytes from healthy volunteers, who gave informed consent, were approved by the Ethics Board of Institut Pasteur (EVOIMMUNOPOP-281297) and the relevant French authorities (CPP, CCITRS and CNIL).

**Samples and dataset.** The high-density genotyping and RNA-sequencing data used in this study were generated as part of the EvoImmunoPop project[38]. Briefly, the EvoImmunoPop cohort is composed of 200 healthy, male donors of self-reported African and European ancestry, all living in Belgium (100 individuals of each population).

Genotyping data was obtained, for all individuals, using both Illumina HumanOmni5-Quad BeadChips and whole-exome sequencing with the Nextera Rapid Capture Expanded Exome kit, leading to a total of 3,782,260 SNPs after stringent quality control. This dataset was then used for imputation, based on the 1,000 Genomes Project imputation reference panel (Phase 1 v3.2010/11/23), leading to a final set of 19,619,457 high-quality SNPs, including 7,650,709 SNPs with a MAF >5% in our cohort. Details on filtering criteria and quality control have been provided elsewhere[38].

RNA was collected from non-stimulated and stimulated monocytes, which were purified from peripheral blood mononuclear cells (PBMCs) with magnetic CD14 microbeads, and sequenced at a depth of ~34 million single-end reads per sample, yielding a final dataset of 970 transcriptional profiles. The purity of the CD14$^+$ monocyte fraction and its lack of contamination by neutrophils were assessed by flow cytometry (Supplementary Note 6; Supplementary Fig. 8). Monocytes from each individual were exposed, for 6 h, to four stimuli including LPS (bacterial lipopolysaccharide, activating TLR4, $n = 184$), Pam$_3$CSK$_4$ (synthetic triacylated lipopeptide, activating TLR1/2, $n = 196$), R848, (imidazoquinoline compound, activating TLR7/8, $n = 191$) and to a human seasonal IAV (strain A/USSR/90/1977 (H1N1), $n = 199$), and compared with resting monocytes from the same individuals ($n = 200$). Normalised gene expression values (FPKM) were computed with CuffDiff and corrected for GC content, ratio of average gene body coverage at 5′/3′ ends of the gene, using linear models and for date of experiment and date of library preparation using ComBat[70]. In addition, *htseq-count* was used to measure, for each sample, the total number of intronic read counts per gene (defining intronic regions as genic regions that are not part of any known exon). Intronic read counts were then divided by total intron length and total read count of the sample, to yield intronic RPKM that were used as a proxy of the transcription rate in downstream analyses.

**Quantification and definition of splice sites.** Reads from all samples were aligned on hg19 genome using the HISAT2 software version 2.0.1[71] and the publicly available *genome_snp* index, which includes all common SNPs from dbSNP 144. Known splice sites were extracted from Ensembl Genes v70 using the *hisat2_extract_splice_sites* script, and used as input to the *–known-splicesite-infile* option to guide the mapping of spliced reads. All other parameters were left to their default value. Spliced reads with an overhang of at least six nucleotides into each exon were then extracted from HISAT2 aligned reads using the *filter_cs* script from leafcutter package[72], and reads spanning over 100 kb, or falling outside genic regions, were discarded to reduce the risk of mapping errors. The resulting set of >7.2 billion spliced reads was used to identify the boundaries of spliced-out introns, and we defined donor and acceptor sites as the two-base pairs located at each end of the detected introns. We then compared the detected splice sites with those of the intropolis database[73], which regroups 42,882,032 introns previously observed in the European Nucleotide Archive, and estimated the frequency of non-GT/AG splice sites as a measure of false positive rate (Supplementary Fig. 1a–c). For all subsequent analyses, we filtered spliced reads overlapping multiple genes, reads mapping low expressed genes (FPKM < 1), and reads for which the corresponding intron is absent from intropolis[73].

To distinguish splice sites that are present in a majority of transcripts from those that are present only in a subset of transcripts, we considered, for each splice site, the splice site to which it is most frequently joined, and extracted all spliced reads that overlap the corresponding intron (i.e., the most frequently spliced intron). We then estimated the percentage of transcripts where the splice site is active as the ratio of the number of reads that support the splice site, to those that overlap the most frequently spliced intron. We defined high-activity splice sites as those supported by an average of one read per sample in at least one condition, and that are used in at least 5% of transcripts. Remaining sites were considered as weakly active (weak splice sites). Weak splice sites that are absent from Ensembl annotations and show no evidence of conservation (GerpRS < 2, see below) were considered as cryptic. Among high-activity splice sites, those that are

used in over 95% of transcripts were considered as constitutive, whereas others were considered as alternative. In addition, splice sites were annotated as coding if they were located next to coding exons in Ensembl v70, and as non-coding otherwise.

**Long-term evolutionary analysis of splice sites**. To measure the conservation of splice sites across mammals, we retrieved pre-computed base-wise GerpRS scores[42] for hg19 from the website of the Sidow Lab (http://mendel.stanford.edu/SidowLab/downloads/gerp/index.html). To reduce the risk of misalignments, we excluded from our analysis positions that (i) had a null GerpRS score, (ii) were absent from the UCSC MULTIZ-46way alignments[74] or (iii) had a negative score in the MULTIZ-46way alignments. We then considered, for each splice site, the mean GerpRS score over the 2 bp that constitute the essential splice site (AG/GT sequence, for canonical splice sites) as a measure of splice site conservation. Gene Ontology (GO) enrichment of genes harbouring non-conserved splice sites was performed with GOSeq[75], adjusting for the total number of active splice sites in the gene and using all genes with at least one active splice site as background set. Enrichments of non-conserved splice sites among genes that are upregulated upon stimulation was assessed using logistic regression. Specifically, we modelled the probability that a given splice site is conserved as a function of the gene's fold-change in expression and maximal level of expression across all stimulations. We then tested for a significant effect of being upregulated ($\log_2FC > 1$) on the probability of being conserved, adjusting on gene expression, with a likelihood-ratio test. The following bins were used for gene expression: [1 < FPKM ≤ 5], [5 < FPKM ≤ 10], [10 < FPKM ≤ 50], [50 < FPKM ≤ 100], [100 < FPKM ≤ 500], [500 < FPKM ≤ 1000], [1000 < FPKM]. Phylogenetic ages of splice sites were estimated from MULTIZ-46way alignments by reconstructing the ancestral sequence and dating the first occurrence of the splice site based on the ancestors of modern humans (Supplementary Note 1).

To test the enrichment of recent splice sites among immune genes—defined based on either GO (immune response term, GO:0006955) or their response to immune stimulation ($\log_2FC > 1$ in at least one condition, with respect to the basal state)—we first estimated the phylogenetic age of each gene as the age of its most ancient splice. We then compared the mean phylogenetic age of splice sites of immune genes with the age of 1000 random sets of splice sites of non-immune genes matched for phylogenetic age and levels of gene expression (using the same bins as previously). When computing $p$-values, a normal distribution was fit to the null distribution obtained from the resampling to increase the precision beyond $10^{-3}$.

**Definition and quantification of AS events**. Human transcripts were retrieved from Biomart (Ensembl Genes 70—GRCh37), and SUPPA *eventGenerator* script was used to annotate local AS events (AS; script available at https://github.com/comprna/SUPPA). This resulted in a total of 163,525 annotated AS events (Alternative first exon—AF: 68,751, Alternative last exon—AL: 17,536, Alternative 3′ splice site—A3: 15,054, Alternative 5′ splice site—A5: 14,274, Mutually exclusive exons—MX: 4,768, Retained intron—RI: 5,789, Skipped exon—SE: 37,353).

To ensure sufficient power in the detection of both isoforms at AS events, we first focused, for each condition of immune stimulation, on genes with an average FPKM > 10, and AS events where each possible alternative intron was supported by at least 30 spliced reads. We then quantified the resulting 39,030 AS events using *MISO* to obtain PSI values[12] and filtered out, in each condition, rare AS events with a mean PSI value under 5% or over 95%, as well as AS events with >5% of missing values (due to the lack of informative reads in the sample). Remaining missing values were imputed by K-nearest neighbours with the *impute.knn* function from the *Impute* R package. As a result of these filters, we obtained a set of 30,796 frequent AS events, for which the less frequent isoform has FPKM > 0.5 in at least one condition and 95% of samples have informative reads. In our setting (~35 M reads per sample) and at this level of expression (FPKM = 0.5), we expect to find ~15 reads per sample for each kb of the minor transcript, providing high power for the detection of both isoforms.

For each AS event, we defined its boundaries as the most upstream and downstream splice site of the alternatively spliced introns. In addition, we extended these boundaries to the most upstream TSS for alternative first exon and the most downstream transcription end site (TES) for alternative last exons. We next computed correlations between PSI value profiles of overlapping AS events, and considered AS events as related events when their PSI values showed an absolute correlation > 0.5. We thus defined a graph of AS events relationships, and used the *cluster_walktrap* function from the *igraph* R package to define clusters of highly connected AS events. We then selected a single representative event for each cluster, based on total counts of spliced reads supporting the events. After filtering overlapping AS events, we obtained a final set of 16,173 AS events for analysis (A3: 2,625, A5: 2,062, AF: 2,018, AL: 1,167, MX: 566, RI: 1,415, SE: 6,320), which we adjusted for batch effects and technical covariates using ComBat[70] (Supplementary Note 2; Supplementary Fig. 2f).

To assess the putative consequences of AS events at the protein level, we retrieved transcript annotations from Ensembl v94 and considered for each AS event, the set of transcripts that are compatible with each possible splicing (reference and alternative isoforms) and the corresponding protein(s). We annotated AS events as *modified protein* when both isoforms are protein-coding, *loss-of-function* if only the reference isoform is protein-coding, *gain-of-function* if

only the alternative isoform is protein-coding, and *non-coding* if both isoforms are non-coding.

**Differential splicing and isoform diversity**. For each annotated AS event, differential splicing between stimulation conditions or populations was assessed using Wilcoxon rank test, and $p$-values were corrected for multiple testing across all conditions using the *'fdr'* method from the *p.adjust* R function (Benjamini–Hochberg FDR correction). When testing differential splicing between conditions, only genes with FPKM > 10, both at basal state and after stimulation, were considered. We considered a change in isoform ratios as significant if it satisfied the following criteria: (i) adjusted $p$-value < 0.05 and (ii) difference in PSI > 0.05. We considered genes as being differentially spliced (between conditions or populations) when they significantly changed their isoform ratios (PSI), for at least one of their AS events. For each gene and condition, isoform diversity was measured by Shannon entropy, assuming that all AS events of a gene occur independently. Shannon entropy ($H$) was computed as follows

$$H\big(\text{gene}_i\big) = \sum_{j \in \text{AS event}} H\big(\Psi_j\big) = \sum_{j \in \text{AS event}} \Big[-\Psi_j \log\big(\Psi_j\big) - \big(1 - \Psi_j\big) \log\big(1 - \Psi_j\big)\Big] \tag{1}$$

where $\Psi_j$ is the mean PSI ratio of the $j$th AS event of gene $i$.

**Quantification of noisy splicing**. To quantify the degree of noisy splicing per gene, we considered, for each constitutive splice site next to a coding exon, the set of all spliced reads that link the splice site to a cryptic splice site (i.e., weak, non-conserved splice site absent from Ensembl annotations) as splicing errors. We then computed, for each condition, the proportion of such splicing errors among all reads that start/end at the splice site, and averaged these values across all constitutive splice sites of a gene to obtain its rate of noisy splicing. In addition, we computed these proportions separately for each sample, and averaged the levels of noisy splicing across all genes, to yield an estimate of the degree of noisy splicing per sample. We assessed the contribution of NMD to noisy splicing by correlating the degree of noisy splicing per sample, with the expression of the key NMD genes *UPF1–3* and *SMG1–9* (summarised by their first PC). For each gene, mean intron length was determined based on the transcript with the largest number of exons among transcripts with FPKM > 1. To assess the impact of transcription rate on noisy splicing, we used the mean intronic RPKM observed at the basal state as a proxy of the transcription rate. We assessed the impact of these features on noisy splicing by Spearman's rank correlation tests, and used Wilcoxon rank test to assess the difference in levels of noisy splicing across conditions. The partial correlation between gene expression and noisy splicing, adjusted for transcription rate, was obtained by regressing ranks of gene expression and noisy splicing at basal state on those of transcription rates, and correlating the residuals.

**Mapping of splice QTLs**. To map genetic variants associated with differences in splicing, i.e., sQTL, we used variants with a MAF ≥ 0.05 in both populations combined, resulting in a set of 7,650,709 SNPs, of which 5,634,819 were located < 1 Mb from an AS event. We mapped local sQTLs, likely *cis*-acting, with *Matrix-EQTL*[76], using a 1 Mb window on each side of the splicing event boundaries. sQTL mapping was performed separately for each condition and merging both populations. PC1 and PC2 of the genotype matrix were included in the model to account for population stratification, and PSI values were rank transformed to a normal distribution before mapping, to reduce the impact of outliers.

FDR was computed by mapping sQTLs on 100 permuted datasets, in which genotypes were randomly permuted within each population. We then kept, for each permutated dataset, the most significant $p$-value per AS event, across all conditions, and computed the FDR associated with various $p$-value thresholds ranging from $10^{-3}$ to $10^{-50}$. We subsequently selected the $p$-value threshold that provided a 5% FDR ($p < 4.8 \times 10^{-7}$). To compare sQTLs across conditions, we used a Bayesian framework to select the most likely model of sQTL sharing across conditions (Supplementary Note 4). sQTLs for which the gene was expressed specifically after stimulation (FPKM < 10 in the non-stimulated state and $\log_2FC > 1$), were annotated as sQTLs of stimulation-specific genes. rsQTLs were defined as those where (i) the gene was expressed at FPKM > 10 both before and after stimulation, (ii) the sQTL was significant specifically after stimulation ($p > 0.001$ in the non-stimulated state) and (iii) the sQTL was significantly associated to changes in PSI value upon stimulation ($p < 0.01$, corresponding to ~5% FDR, when testing all sQTLs across the 4 stimuli).

**Regulatory elements and predicted impact on splicing**. The SPIDEX[TM] database[36] was retrieved from the deepgenomics website (https://www.deepgenomics.com/spidex-noncommercial-download), and functionality scores (Z index) were extracted from all SNPs present in the database (i.e., located < 300 nucleotides apart of a known splice site). Scored SNPs were annotated as exonic or intronic based on Ensembl v70 annotation, and considered deleterious when their absolute Z-score was above 2, and benign otherwise. We then used Fisher's exact test to

assess the enrichment of sQTLs in exonic and intronic regions, as well as in predicted splice-altering regions, considering the peak SNP at each locus.

To further determine the set of regulatory elements that contribute to regulate splicing, we annotated SNPs for various types of regulatory features including overlap with (i) donor/acceptor sites (2-bp upstream or downstream of each exon), (ii) splice site flanking regions (3–8-bp upstream or downstream of an exon), (iii) expected branchpoint regions (18–39-bp upstream of acceptor site), (iv) transcriptional regulatory elements (promoter, promoter flanking, enhancer, and CTCF binding sites) that are active in monocytes, based on Ensembl Regulatory Build v80[77] and (v) conserved regions (GerpRS > 2). In addition, to explore how sQTLs overlap motifs of RNA-binding proteins, we retrieved a set of 107 motifs bound, in vitro, by 81 human splicing factors and RNA-binding proteins[78]. We then kept a single motif per protein, giving priority to the motif with the highest information content, and used Homer[79] to scan the reference genome sequence for these motifs. Namely, we used *findMotifsGenome.pl* script with -mask and -rna to identify motifs of human RNA-binding proteins in a 20 nucleotides window around each SNP (considering both strands), and excluded motifs that did not directly overlap the SNP or presented a motifScore of 5 or less. We used a Fisher's exact test to assess the enrichment of sQTLs in SNPs that overlap the regulatory motifs and features described above, with respect to genome-wide expectations. For all analyses, SNPs with a MAF ≥ 0.05 and located < 1 Mb away from an AS event were used as background set.

**sQTL/eQTL overlap and causality inference**. To measure the impact of sQTLs on gene expression, we also mapped eQTLs from rank-transformed gene-level FPKM values, with MatrixEQTL[76]. Local eQTLs were mapped in a 1 Mb window around each gene, merging the two populations and adjusting for PC1 and PC2 of the genotype matrix. FDR was computed through permutations as performed for sQTLs. To identify shared genetic control of splicing and gene expression, we considered for each sQTL, the levels of LD ($r^2$), computed across both populations, between the sQTLs and eQTLs detected for the same gene and condition, as a measure of the colocalization between the genetic determinants of splicing and gene expression. Causal relationships between splicing and expression changes observed at sQTLs were assessed using a two-step approach and a modified version of the Likelihood-based Causal Model Selection Framework described in[44] (Supplementary Note 3; Supplementary Fig. 6). To assess the impact of sQTLs on transcription rate, we tested, for each sQTL, the effect of the peak sQTL SNP on intronic RPKM in the condition where the sQTL was the most significant, adjusting for population of origin. sQTLs associated to intronic RPKM with $p < 0.05$ were considered as significantly associated with transcription rate.

**Overlap of sQTLs with GWAS loci**. We downloaded the NHGRI GWAS catalogue[46] from EBI (date: 26 June 2017), and selected all SNPs that were significantly associated with a complex trait or disease at $p$-value of $10^{-5}$ ($N = 27,884$). GWAS traits were annotated based on the EFO mappings[47], and grouped according to their parental EFO categories. Note that a single locus/trait can be assigned to multiple EFO categories. To account for uncertainty of GWAS mapping, we extended the set of GWAS SNPs to all SNPs in LD with a GWAS SNP in Europeans ($r^2 > 0.8$), given that most GWAS are performed on individuals of European ancestry. Using this criterion, we estimate that 5.7% of all SNPs considered in our analyses are associated with at least one trait, with no >0.2% of the tested SNPs being associated to a single phenotype (e.g., height). After excluding nonspecific EFO categories (i.e., other disease, other trait, other measurement and biological process), we obtained a final set of 229,425 SNPs associated with at least one of the 14 EFO terms (4.1% of all tested SNPs).

For enrichment analyses, we removed sQTLs that were in LD ($r^2 > 0.8$) to keep a single SNP per haplotype (1,271 independent SNPs) and compared, for each EFO category, the number of sQTLs overlapping GWAS loci, with respect to genome-wide expectations: all SNPs initially used for sQTL mapping, pruning them for LD ($r^2 > 0.8$), were used as background set (568,973 SNPs). For each EFO category, 1000 resamples were used, matching for bins of MAF at intervals of 2%. Enrichment $p$-values were calculated by counting the frequency at which the number of GWAS loci in the 1000 resampled sets of SNPs, exceeded the number of GWAS loci observed in our data. For the example of the *TMBIM1* sQTL, we downloaded inflammatory bowel disease (IBD) summary statistics from https://www.ibdgenetics.org/downloads.html, and extracted GWAS $p$-values from the European-based GWAS meta-analysis. The *TMBIM1* sQTL was remapped in European individuals only, to allow comparison with the GWAS $p$-values.

**Population differences in splicing and mediation analysis**. For genes that are differentially spliced between populations and harbour a sQTL, we evaluated the fraction of population differences in splicing that is attributable to the sQTL, using the *mediate* function from the *mediation* R package[57]. Briefly, mediation analyses decompose the population differences in isoform ratios between the effect that is due to differences in allelic frequency at the sQTL (mediated or indirect effect, $\zeta$) and the effect of population that occur independently of the SNP under

consideration (independent or direct effect $\delta$).

$$\Delta PSI = E(PSI|AFB) - E(PSI|EUB) = \zeta + \delta \quad (2)$$

with

$$\zeta = E(PSI|E(snp|AFB), pop) - E(PSI|E(snp|EUB), pop) \quad (3)$$

and

$$\delta = E(PSI|snp, AFB) - E(PSI|snp, EUB) \quad (4)$$

Under a linear model, the mediated effect can be expressed as:

$$\zeta = 2\beta(f_{AFB} - f_{EUB}) \quad (5)$$

where $f_{AFB}$ and $f_{EUB}$ are the allelic frequency of the SNP under consideration, and $\beta$ is the impact of a single allele of the sQTL on PSI values.

The proportion mediated $\tau$ can then be estimated as:

$$\tau = \frac{\zeta}{\zeta + \delta} = \frac{2\beta(f_{AFB} - f_{EUB})}{\Delta PSI} \quad (6)$$

**Detection of signals of positive selection at sQTLs**. To detect signatures of population-specific natural selection, we used two metrics, $F_{ST}$ and iHS, which detect signals of local adaptation. $F_{ST}$ quantifies the variance of allele frequencies within and between populations to detect outlier values of population differentiation[58], which may result from the action of positive selection in one specific population. The iHS[59] compares the degree of extended haplotype homozygosity of the derived and ancestral alleles, allowing to identify differences in haplotype length between alleles that can result from the rapid increase in frequency of the putatively selected allele.

To identify candidate sQTLs under selection, we used an outlier approach by computing the top 5% values of $F_{ST}$ and |iHS| at the genome-wide level, in each population separately. To support further the adaptive nature of candidate sQTLs, we tested for local enrichments in outliers of $F_{ST}$ or |iHS| (top 5% of signals) within a 100 kb window around each sQTL (focusing on SNPs with MAF ≥ 0.05), as previously performed[38,59,60]. Indeed, it has been shown that searching for the occurrence of local enrichments in outliers increases the rate of true selected loci among genome-wide outliers[59]. We compared the number of outliers within each window with a beta-binomial distribution with parameters ($\alpha$, $\beta$, $N$), where $N$ is the number of informative SNPs (MAF ≥ 0.05) in the window, and the parameters ($\alpha$, $\beta$) are obtained by fitting a beta distribution to the proportion of outliers of $F_{ST}$ or |iHS| per 100 kb window genome-wide.

**Detection of Neanderthal-introgressed sQTLs**. To measure the effect of admixture with Neanderthals on splicing, we used the complete genome sequence of the Neanderthal Altai[80] to identify a total of 100,755 frequent SNPs (MAF > 5%) that are exclusively present on Neanderthal haplotypes in Europe (aSNPs, Supplementary Note 5). We then counted the number of independent loci where the peak SNP is either an aSNP or tags an aSNP in Europe ($r^2 > 0.8$), and compared this number with genome-wide expectations by resampling. All SNPs initially used for sQTL mapping, with a MAF > 5% in Europe, were LD pruned ($r^2 > 0.8$) and used as background set (568,973 SNPs). We then randomly resampled 1000 sets of independent SNPs matched for allelic frequencies with sQTLs, and counted the number of tag-aSNPs in each set. Enrichment $p$-values were calculated by counting the frequency at which the number of tag-aSNPs in our 1000 resampled sets of SNPs exceeded the number of tag-aSNPs observed in our data. To assess the robustness of our enrichment analyses to changes in the definition of Neanderthal haplotypes, we repeated the resampling with more stringent definitions of archaic haplotypes (i.e., increasing the haplotype length and number of Neanderthal alleles required to call a haplotype as archaic, Supplementary Note 5).

**Reporting summary**. Further information on experimental design is available in the Nature Research Reporting Summary linked to this article.

## Data availability

Genome-wide SNP genotyping, whole-exome sequencing and RNA-sequencing data used in this study have been deposited in the European Genome-phenome Archive (EGA) under accession code EGA: EGAS00001001895. Source data for figure(s) 1–6 are provided with the paper. All other relevant data are available upon request.

## Code availability

All scripts have been deposited on github under MIT License (https://github.com/mrotival/EvoImmunoPop_Splicing).

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

## Acknowledgements

This project was funded by the European Research Council under the European Union's Seventh Framework Programme (FP/2007–2013)/ERC grant agreement 281297 (to L.Q.-M.). The Quintana-Murci lab is funded by the French Government's Investissement d'Avenir program, Laboratoires d'Excellence "Integrative Biology of Emerging Infectious Diseases" (ANR-10-LABX-62-IBEID) and the Fondation pour la Recherche Médicale (Equipe FRM DEQ20180339214). M.R. was supported by a Marie Skłodowska-Curie fellowship (DLV-655417). We thank Etienne Patin and Mary O'Neill for helpful comments and suggestions.

## Author contributions

M.R. designed and performed the computational analyses, and analysed the data. M.R., H.Q. and L.Q.-M. interpreted results and wrote the paper. L.Q.-M. supervised the research and obtained the funding.

## Additional information

**Competing interests:** The authors declare no competing interests.

