## [Peer Review File · Nature Communications]

Reviewers' comments:

Reviewer #1 (Remarks to the Author):

In their submitted manuscript, "Defining the genetic and evolutionary architecture of alternative splicing in response to infection", Rotival et al. contrast the splicing landscape of resting and stimulated monocytes from 200 individuals of African and European ancestry. They characterize ~2 million exon-exon junctions identified by RNA-sequencing, classifying the corresponding splice sites as weak or high-activity (the latter in turn composed of alternative and constitutive splice sites).

The authors investigate the distribution of associated GERP scores to gain insight into the evolutionary constraints on splicing, finding that constitutive and alternative splice sites tend to exhibit relatively high conservation, while the much larger class of weak splice sites tend to exhibit low conservation and are enriched in immune response genes. They showed that splice sites of immune genes tend to be younger--including several human-specific (relative to non-human primate) splice sites that they identified. They interpret this finding to support rapid diversifying selection which has previously been demonstrated to disproportionately affect immune-related genes.

The authors also conduct several variations on analyses aimed at understanding how alternative splicing responds to infection. For example, they show that isoform diversity increases with immune challenge, but that this is mostly driven by an increase in expression of non-functional isoforms (presumably due to downregulation of NMD pathways). To understand the impact of genotype on splicing, the authors test for local (cis-) splicing quantitative trait loci (sQTLs) and reassuringly show that most significant cis-sQTLs lie very close to the AS events they control. They extend this analysis to the identification of GxE interactions (i.e., where the magnitude/direction of effect of an sQTL depends on resting vs. stimulated condition). Finally, the authors intersect their sQTLs with data from GWAS, selection scans, and maps of Neandertal ancestry to better understand how immune-associated splicing contributes to complex phenotypes and is shaped by human evolution.

I found the manuscript to be very interesting and well written and most of the analyses to be careful and convincing--generally supported by multiple lines of evidence. While none of the results were particularly unexpected, this work makes a strong case that genetic variation influencing alternative splicing is a major contributor to phenotypic diversity. Investigating this topic in the context of immune challenge sheds important light on the prevalence of gene by environment interactions. Moreover, immunity is of special interest from an evolutionary standpoint, as arms races with pathogens may have exerted some of the strongest selective pressures in during human evolutionary history.

Major comments:

- 1) Analysis code/scripts should be posted to GitHub, uploaded as supplementary materials, or deposited in some more permanent archive. This step supports reproducibility and transparency.
- 2) The Likelihood-based Causal Model Selection Framework employed by the authors is very interesting and potentially powerful for disentangling the direction of causation. I am not familiar with this approach, however, and am surprised to learn that this would be possible given the structure of your data. I think it would be valuable to add a few sentences providing a high-level description of how such inferences can be achieved. How are the likelihoods calculated in practice? It was also non-intuitive to me that the change in splicing was frequently driven by changes in gene expression rather than vice versa. Are there case examples of such a model in the literature? If so, they may be worth citing here.

3) The analyses of overlaps with GWAS hits and signatures of historical positive selection were not particularly convincing. In the former case, we have no way of knowing whether the sQTL is causal in the association or simply in LD with another causal variant. The recent paper presenting the omnigenic model (Boyle et al, 2017, Cell), for example, showed that most SNPs (>60%) are in LD with a variant that influences height—and that is just a single phenotype. The enrichment signal observed in the current study is somewhat more convincing, but I am worried that 5% MAF-matching could be too liberal. How do results change with a stricter threshold? In the case of the selection scans, it should be noted that any distribution (F_{st} , iHS , etc.) will have outliers. So I am skeptical about how much confidence should be placed in these anecdotes (e.g. RABGAP1, CD226) given that we don't know where to set the threshold for identifying true cases of positive .

4) The enrichment of Neanderthal SNPs for sQTL was very modest. I would therefore encourage the authors to interpret the result cautiously. How does it change when you alter your thresholds for defining the original set of Neanderthal-introgressed SNPs?

Minor comments:

1) Overall, the figures are quite visually appealing. I have a couple of suggestions to simplify them, however, for easier interpretation:

- Fig. 1a: semi-transparent histograms are difficult to interpret due to the overlapping bars. Can this be changed to a density plot? Or three histograms stacked vertically with a shared X-axis?

- Fig. 1b/c: contain redundant information. I think it would be preferable to eliminate one of these panels for simplicity.

- Fig. 1d: before diving into the methods, it seemed strange to me that “age” would not be given in units of time, but by taxonomy. Can this legend label be changed for accuracy?

- Fig. 4a: what exactly is being depicted by the schematic at the top? Is it necessary, and can it be removed?

2) I would advise avoiding the term “maladaptive” toward the end of the discussion. Pleiotropy is widespread, and we cannot be sure that the phenotype you have identified as showing association was the target of selection during human evolutionary history.

Reviewer #2 (Remarks to the Author):

Rotival et al composed a well written article that extends the work from their previous study (Quach 2016). Previously, the authors isolated CD14 positive monocytes from 200 individuals (100 African and 100 European) and analyzed correlations between genotype and gene expression (eQTL). The monocyte transcriptomes were analyzed in the non-stimulated state and in response to four other treatments. The authors' previous article expanded our understanding of the interplay between human genetic variation/evolution with transcriptional responses to immune challenges.

In the current article, they authors used the same data and analyzed splicing (sQTL). No new primary datasets were generated for this article and much of the general analysis/results are quite similar to their previous publication.

Major comments:

1. A major potential issue for this work is how to decouple the effects of gene expression and alternative splicing, and to control for biases that arise from this coupling. By definition, a gene non-expressed in the basal state but activated upon immune stimulation would never have a detectable sQTL signal in the basal state. Likewise, such a gene is more likely to have minor (and cryptic) splicing events detected in the activated state, given the increase in total gene expression upon immune stimulation. I worry that various statements and conclusions in this paper could be attributed to a bias for detecting regulated or cryptic splicing events when gene expression levels are higher (thus better power by RNA-seq). This issue needs to be examined and properly controlled for in a revised manuscript.

2. The definition of condition-specific sQTLs may also need scrutiny and better rigor. We should not define condition-specific sQTLs simply as events detected as significant in one condition but not another: this could be due to statistical fluctuation or differential power in different conditions (e.g. the gene is more highly expressed in one condition so more RNA-seq coverage and better power for that condition). To define condition-specific sQTLs more rigorously we should look at effect size of genotypes on splicing in different conditions.

3. I have several concerns with the section on noisy splicing. Here, the authors define a metric for splicing noise and use this to quantify splicing noise across the five conditions. They quantify this value for all genes and conclude that noisy splicing increases after stimulation but is maintained at low levels for immune genes.

My main issue with the noisy splicing section is with the mathematical definition of noisy splicing. For all constitutive splice sites in a given gene, the proportion of correctly spliced reads are multiplied together and the complement of this product is used as a measure of noisy splicing. As a mathematical consequence of this definition, a gene with many introns would have noisier splicing than a gene with few introns. Indeed, the authors find a positive correlation between noisy splicing and intron number (page 10 line 6-7). However, the authors imply that this correlation is a biological phenomenon rather than a mathematical artifact.

It seems like the authors' rationale for multiplying the proportion of correctly spliced reads is to assume random assortment of noisy splicing and calculate the proportion of transcripts that have no incorrectly spliced introns. With this definition of noisy splicing, the values are not comparable across genes. However, the authors aggregate this measurement across all genes within the five conditions to come to their conclusions. Since the measurement is not comparable between different genes, and gene expression levels change across the five conditions, the authors' conclusions may be invalidated if they restrict their analysis individually within genes.

I would recommend that the authors restrict the noisy splicing analysis to the splice site level or pool the reads across all constitutive splice sites within a given gene to get an average value and compare these values individually within genes across the five conditions to see if they come to the same conclusions. The latter part would be necessary to do an "apples to apples" comparison between different conditions.

4. The authors compare putative regulatory variants that are shared between the sQTL and eQTL analysis and infer causality between SNP, expression, and splicing. However, readers may question the accuracy of these causal inferences, given the limitations of the data. It may help persuade readers to include specific examples of each type of causal model. For example, an sQTL in ERAP2 is

known to introduce a premature termination codon and trigger nonsense mediated decay, so differential expression is causally impacted by differential splicing in this case.

5. For the section on regulatory elements and predicted impact on splicing, the four categories of regulatory elements seem to pertain more towards gene expression rather than splicing. RBP binding sites, RNA secondary structure, and conserved sites might be better candidates to look at.

Minor comments:

Figure 1d: I would recommend reversing the order of the estimated age legend so that the order of the color scale is in the same orientation with the data.

Additionally, the brackets indicate that three types of comparisons were done, but it is a bit confusing why those specific sets of comparisons (but not others) were done.

Also, the legend defines a two asterisk (**), but there is no (**) in the plot.

Figure 2c: It is a bit confusing what the brackets refer to and why there are only three sets of asterisks. Shouldn't there be four sets of comparisons (i.e. the four treatments to NS)?

Figure 3b: The labels on the x-axis seems a bit strange. The last tick mark is labeled as ">20" but the preceding tick mark is "20". Should the last tick mark be changed to ">=30"?

Figure 3d: The NS plot should be added here for completeness.

Figure 4fg: These plots show the isoform expression levels, but a plot of PSI values would help emphasize the shift in alternative splicing (Similar to Figure 2e).

Figure 6b: The endpoints of intervals are not included in any of the marked intervals. i.e. if $\beta = 0.05$, is it included in the plot? Please modify this figure for accuracy.

Reviewer #3 (Remarks to the Author):

NCOMMS-18-18589: Defining the genetic and evolutionary architecture of alternative splicing in response to infection" by Prof Quintana-Murci and colleagues

Immune stimulation of monocytes is shown to increase mRNA isoform diversity through alternative splicing (AS). Perhaps surprisingly, a negative correlation between AS and gene expression is reported, suggesting that gene expression down-regulation correlates with an increase in AS. Finally, results are presented indicating that African- or European-specific alternative splicing patterns can be detected and that eight loci showing AS can be traced back to genetic Neanderthal admixture in Europeans.

This work is based on a RNAseq data set published in Cell in 2016 by the same research group (PMID:27768888) where AS was not considered. The research is an example of data re-analysis which may inspire others in the field. Surprising results emerged, namely the negative correlation of AS and RNA level reduction which may indicate hitherto unsuspected or not (yet) well documented relation between transcription frequency and splicing efficiency. The authors speculate that the increased detection of AS on lowly expressed genes may reflect a modulation of nonsense mediated mRNA decay (NMD). If this is the case, a stimulus-specific pattern of AS of lowly expressed genes would be expected. However I did not find this back in the manuscript? One alternative and radical interpretation a reader of this manuscript might come away with is that it may be possible that cells not only reduce the transcription of specific genes but also further inhibit specific gene expression levels by actively 'scrambling' RNAs through AS events. Hence the AS events qualified as 'noisy' by the

authors could (highly speculatively) be an active gene expression downregulation mechanism. However, at this stage it is also possible that the increased AS detected at repressed genes is an artefact of the analysis pipelines. From the text going from P6L7 to P6L15, it is not obvious whether/how or not the analysis pipeline takes into account to what level an RNA is present (expressed as RPKM or FPKM) to call an AS. This could be explicitly stated if RNA level is not part of the filtering procedure.

AS as observed by the authors can be split into two categories: (i) AS that concerns RNAs that are well expressed and that will result in alternative protein isoforms being produced constitutively or under specific circumstances (unstimulated, LPS stimulation, virus exposure) and (ii) AS of (very) weakly expressed genes that show noisy splicing patterns. While the former is of general interest (human variation and evolution, immunology, RNA and protein research), the latter may reflect quixotic RNA metabolism or the coupling of splicing efficiency to transcription frequency, a field of investigation with many unresolved if not contentious questions. One weakness of the manuscript as it stands is that the authors do not distinguish well between (i) and (ii) in some sections of the results and discussion sections. This can be improved substantially, in particular at P9L20-22, p10L19-21, P14L21 (are these sQTLs associated with gene induction or gene repression?), P18L3 (does 'spliced' refer to AS, or to the ratio of intronic versus exonic reads?), P18L5 (are the 993 genes more or less alternatively spliced upon induction, or is there no quantitative bias in the isoforms produced and thus simply alternative AS? – as it stands this is a number but it does not really distinguish between (i) and (ii) above?)

Major comments:

1) It is known that neutrophil transcriptomes are riddled with partially spliced RNAs (Orchestrated Intron Retention Regulates Normal Granulocyte Differentiation - PMID:23911323 (2013), Genetic Drivers of Epigenetic and Transcriptional Variation in Human Immune Cells - PMID:27863251 (2016)). It is also known that neutrophils are purified by CD14 beads (PMID:27863251). Usually the neutrophils are lost and monocytes are enriched upon selection for adhesion to Petri dishes. It is unclear to me whether the monocyte purification, conservation and stimulation protocols used (as published by the authors of the present study in Cell in 2016 PMID:27768888) preserved this neutrophil sub-population. Since a major part of the alternative splicing events concerns genes expressed at a low level (see comment xx), a concern is that the authors 'scraped the bottom of the barrel' in their search for alternative splicing events and ended up studying contaminating neutrophil RNAs rather than the monocyte RNAs they intended to study in the first place. One concrete way to check for neutrophil contamination would be to quantify the level of neutrophil-restricted/enriched RNAs such as FCGR3B and SLC44A2 in the 197 transcriptomes and in the AS events documented in the present manuscript. A second approach would be to perform a repeat of the major analyses of the paper exclusively on genes expressed better (> 10-fold?) by monocytes than neutrophils.

2) Have the authors thought to deposit all the new alternative splice sites to a relevant RNA data base? Have the data been shared with Phantom? Alternatively, the Sequence Ontology may provide a repository for the splice sites identified here?

3) It is possible to classify genes as a function of the exon/intron read signal, by mapping RNAseq reads to the genome, or only to cDNAs and computing a ratio of signal using the two approaches. I wonder whether the genes that are expressed at low levels and that show AS are also genes which belong to one end of this spectrum, namely with as many or even more intronic reads than exonic ones. This may help determine whether the source of variation in AS, as defined here, concerns nuclear RNAs that will most likely not give rise to proteins versus exported mature RNAs with a corresponding functional protein translation product.

Minor comments:

Page7 Line12: the authors probably mean $|\log_2FC| > 1$ rather than $\log_2FC > 1$?

P10L21: " due to a reduced rate of noisy splicing amongst highly expressed genes

P11L4: "Most sQTLs (84%) were located within 10 kb of the AS event they control" could be toned down to "Most sQTLs (84%) were located within 10 kb of the respective AS event, with 36% ..."

P11L14: The authors mention "... when excluding the HLA region." What proportion of the 36% of sQTLs falling directly within the boundaries of the AS event (P11L4) are concentrated in the HLA? How much of the 84% mentioned on P11L4 fall in the HLA region? One of the answers to these questions could be included in the text on P11L4.

P12L4-6: "These analyses indicate that the genetic determinants of isoform usage and gene expression are largely independent, yet, when they are not, splicing appears to be mainly driven by changes in gene expression" would be more accurate as "These analyses indicate that the genetic determinants of isoform usage and gene expression are largely independent, yet, when they are not, AS appears to be mainly driven by reduced gene expression"?

P13L9: what are 'body measurements'? Can a literature reference to this trait be included? Is it body size/girth weight? Or ???

P13L19: "Focussing on variants that ..." could be easier to catch if their number (16173??) were mentioned.

P16L3: I would delete 'finally' from this sentence.

Colin Logie

RESPONSES TO REVIEWERS' COMMENTS

Reviewer #1 (Remarks to the Author):

In their submitted manuscript, "Defining the genetic and evolutionary architecture of alternative splicing in response to infection", Rotival et al. contrast the splicing landscape of resting and stimulated monocytes from 200 individuals of African and European ancestry. They characterize ~2 million exon-exon junctions identified by RNA-sequencing, classifying the corresponding splice sites as weak or high-activity (the latter in turn composed of alternative and constitutive splice sites).

The authors investigate the distribution of associated GERP scores to gain insight into the evolutionary constraints on splicing, finding that constitutive and alternative splice sites tend to exhibit relatively high conservation, while the much larger class of weak splice sites tend to exhibit low conservation and are enriched in immune response genes. They showed that splice sites of immune genes tend to be younger--including several human-specific (relative to non-human primate) splice sites that they identified. They interpret this finding to support rapid diversifying selection which has previously been demonstrated to disproportionately affect immune-related genes.

The authors also conduct several variations on analyses aimed at understanding how alternative splicing responds to infection. For example, they show that isoform diversity increases with immune challenge, but that this is mostly driven by an increase in expression of non-functional isoforms (presumably due to downregulation of NMD pathways). To understand the impact of genotype on splicing, the authors test for local (cis-) splicing quantitative trait loci (sQTLs) and reassuringly show that most significant cis-sQTLs lie very close to the AS events they control. They extend this analysis to the identification of GxE interactions (i.e., where the magnitude/direction of effect of an sQTL depends on resting vs. stimulated condition). Finally, the authors intersect their sQTLs with data from GWAS, selection scans, and maps of Neandertal ancestry to better understand how immune-associated splicing contributes to complex phenotypes and is shaped by human evolution.

I found the manuscript to be very interesting and well written and most of the analyses to be careful and convincing--generally supported by multiple lines of evidence. While none of the results were particularly unexpected, this work makes a strong case that genetic variation influencing alternative splicing is a major contributor to phenotypic diversity. Investigating this topic in the context of immune challenge sheds important light on the prevalence of gene by environment interactions. Moreover, immunity is of special interest from an evolutionary standpoint, as arms races with pathogens may have exerted some of the strongest selective pressures in during human evolutionary history.

RESPONSE: We thank the reviewer for their constructive and helpful comments. The answers to each specific point are provided below, as well as the changes that have been made in the manuscript. Overall the manuscript has been substantially revised both in terms of writing and additional analyses (i.e. new results, updated figures and tables, and new supplementary figures), with the main conclusions unchanged but strengthened.

Major comments:

- 1) Analysis code/scripts should be posted to GitHub, uploaded as supplementary materials, or deposited in some more permanent archive. This step supports reproducibility and transparency.*

RESPONSE: We entirely agree with the reviewer. We have now deposited all scripts relevant to the analyses performed in the current manuscript in a GitHub archive. See https://github.com/mrotival/EvolImmunoPop_Splicing.

- 2) The Likelihood-based Causal Model Selection Framework employed by the authors is very interesting and potentially powerful for disentangling the direction of causation. I am not familiar with this approach, however, and am surprised to learn that this would be possible*

given the structure of your data. I think it would be valuable to add a few sentences providing a high-level description of how such inferences can be achieved. How are the likelihoods calculated in practice? It was also non-intuitive to me that the change in splicing was frequently driven by changes in gene expression rather than vice versa. Are there case examples of such a model in the literature? If so, they may be worth citing here.

RESPONSE: We thank the reviewer for their interest in the approach, and we acknowledge that the results, as they were presented, were somehow counter-intuitive. They are several well-documented mechanisms through which changes in transcription could alter splicing (see for example references [1-5]). The presence of a secondary transcription start site can lead to alternative first exons, or, otherwise, an increased rate of transcriptional elongation can lead to reduced recruitment of splicing factors at weak splice sites and thus modulate alternative splicing.

To address the reviewer's concerns, we have now extensively revised this part of the manuscript. In particular, (i) we now explicitly present the possible mechanisms through which splicing could regulate gene expression, or vice versa, and provide specific examples for each situation (Results, p12 ln12-16 and p12 ln24 to p13 ln8), and (ii) we now explain the general principle of causality inferences in the main text (p12 ln17-20), as well as extend the Supplementary Note 3 (supplemented with a new Supplementary Figure 6) to provide an in-depth explanation of the principles and details on the computation of the likelihoods used for the causality inference.

In addition, we have now improved our causality inference analyses to account for cases where gene expression is regulated through both splicing-dependent and splicing-independent mechanisms, as well as for cases where splicing is regulated through both transcription-dependent and independent mechanisms. To do so, we used a two-step approach where we first test for the existence of a causal link between splicing and gene expression, before assessing the most likely direction of this link (see Results, p12 ln17-20 and Supplementary Note 3). Furthermore, we have conducted new analyses to validate our causality inference, through the analysis of intronic reads that emanate from nascent, unspliced mRNAs (as in reference [6]). We reasoned that if gene expression is regulated through alternative splicing, we expect no change in transcription rate between the two alleles of an sQTL. Conversely, we expect to find a significant difference in the amount of pre-mRNA between the two alleles in the opposite situation. Our analyses strongly support these predictions, by showing, for example, that transcription rate differs between the two alleles in 89% of cases where we predict that splicing is driven by gene expression (see p13 ln12-19).

All these results are now presented in the section «Disentangling the genetic regulation of isoform usage and gene expression»

3) *The analyses of overlaps with GWAS hits and signatures of historical positive selection were not particularly convincing. In the former case, we have no way of knowing whether the sQTL is causal in the association or simply in LD with another causal variant. The recent paper presenting the omnigenic model (Boyle et al, 2017, Cell), for example, showed that most SNPs (>60%) are in LD with a variant that influences height—and that is just a single phenotype. The enrichment signal observed in the current study is somewhat more convincing, but I am worried that 5% MAF-matching could be too liberal. How do results change with a stricter threshold? In the case of the selection scans, it should be noted that any distribution (F_{st} , iHS , etc.) will have outliers. So I am skeptical about how much confidence should be placed in these anecdotes (e.g. RABGAP1, CD226) given that we don't know where to set the threshold for identifying true cases of positive.*

RESPONSE: We agree with the reviewer that LD could lead to spurious associations between sQTLs and GWAS signals. However, we used a very conservative threshold to consider variants as linked ($r^2 > 0.8$), which should minimize this risk. Indeed, only 5.7% of common variants in the genome are in LD with a GWAS trait when using our definition. Note that while Boyle et al. report that “62% of SNPs are associated with a non-zero effect with height”, we find that only 0.2% of variants are linked with a variant that influences height when we apply our stringent definition of LD. Thus, our definition of LD should guarantee a strong colocalization between the detected sQTLs and the peak of the GWAS signals, as attested by the case we now present showing the colocalization between a *TMBIM1* sQTL and a GWAS signal of inflammatory bowel disease (see new Fig. 5b). This rationale is now presented in the Methods section (see p31 ln21-23). Furthermore, following the reviewer's request, we have

repeated the enrichment analysis using a more stringent matching on MAF (2% bins), which has not affected the enrichment results (see updated Fig. 5a).

Regarding the selection analyses, while we agree that the presence of outliers does not necessarily mean the occurrence of selection, all candidate sQTLs reported not only fall in the 95th percentile of extreme F_{ST} /iHS values, but, importantly, they are located in regions that present a local enrichment in such outliers, thereby increasing the likelihood of a true selection event. Furthermore, note that all loci discussed in the main text fall in the 99th percentile of F_{ST} or iHS, and that some of them, such as the rs208293 variant regulating *P2RX7*, present strong values of both iHS and F_{ST} , reinforcing the notion of adaptation acting at these loci. To account for the reviewer's comment, we have now revised the text to provide a more objective rationale for the choice of the loci we discuss, and add some level of caution (see Results section, p16 ln24 to p17 ln6 and p17 ln9-14).

The enrichment of Neanderthal SNPs for sQTL was very modest. I would therefore encourage the authors to interpret the result cautiously. How does it change when you alter your thresholds for defining the original set of Neanderthal-introgressed SNPs?

RESPONSE: We thank the reviewer for this question. The main reason why the enrichment was so modest is that we had been overly-conservative in our initial analyses, as we allowed haplotypes subject to incomplete lineage sorting (ILS) in the resampled sets we used to define significance but not in our list of archaic sQTLs. We have now improved our resampling strategy to filter ILS variants *a priori* by computing haplotype length in the exact same way for our resampled sets and observed data (i.e. based on the maximal distance between linked archaic variants). As a result, our conclusions have been strengthened and the reported *p*-value is more significant ($p=0.007$).

To assess the impact of the thresholds used to define introgressed SNPs on our results, we have repeated our enrichment analyses varying the number of aSNPs and physical length required to define archaic haplotypes. The enrichments remained unchanged (see Table 1 below, which is included as Supplementary Table 4f).

Table 1: Enrichment of sQTL in archaic haplotypes, according to the definition of archaic haplotypes

Number of archaic-like SNP on the haplotype	length of the haplotype	Number of sQTL detected	Number of sQTL expected	Fold enrichment	Pvalue
2	10 kb	8	2.84	2.82	0.007
2	20 kb	8	2.07	3.87	0.002
2	30 kb	6	1.50	4.01	0.006
2	50 kb	4	0.96	4.17	0.015
5	10 kb	8	2.48	3.23	0.003
5	20 kb	8	1.94	4.12	<0.001
5	30 kb	6	1.45	4.15	0.005
5	50 kb	4	0.95	4.21	0.015
10	10 kb	8	1.61	4.96	<0.001
10	20 kb	6	1.46	4.10	0.003
10	30 kb	5	1.23	4.07	0.007
10	50 kb	3	0.89	3.37	0.057

Minor comments:

1) Overall, the figures are quite visually appealing. I have a couple of suggestions to simplify them, however, for easier interpretation:

- Fig. 1a: semi-transparent histograms are difficult to interpret due to the overlapping bars. Can this be changed to a density plot? Or three histograms stacked vertically with a shared X-axis?

RESPONSE: We have replaced the histogram of Fig. 1a by a density plot.

- Fig. 1b/c: contain redundant information. I think it would be preferable to eliminate one of

these panels for simplicity.

RESPONSE: We have now removed the Fig. 1b panel. We also replaced Fig. 1c by a density plot, for easier comparison with Fig. 1a.

- Fig. 1d: before diving into the methods, it seemed strange to me that “age” would not be given in units of time, but by taxonomy. Can this legend label be changed for accuracy?

RESPONSE: We now report the taxon name with the inferred age of the most recent common ancestor of the taxon in millions of years.

- Fig. 4a: what exactly is being depicted by the schematic at the top? Is it necessary, and can it be removed?

RESPONSE: The illustration depicted the definition retained for splicing event boundaries in the case of exon skipping. We have now removed the illustration to avoid confusion.

2) I would advise avoiding the term “maladaptive” toward the end of the discussion. Pleiotropy is widespread, and we cannot be sure that the phenotype you have identified as showing association was the target of selection during human evolutionary history.

RESPONSE: The reviewer is completely right. We have now tuned down this part of the discussion accordingly (see p21 ln7-8,12).

Reviewer #2 (Remarks to the Author):

Rotival et al composed a well written article that extends the work from their previous study (Quach 2016). Previously, the authors isolated CD14 positive monocytes from 200 individuals (100 African and 100 European) and analyzed correlations between genotype and gene expression (eQTL). The monocyte transcriptomes were analyzed in the non-stimulated state and in response to four other treatments. The authors' previous article expanded our understanding of the interplay between human genetic variation/evolution with transcriptional responses to immune challenges.

In the current article, they authors used the same data and analyzed splicing (sQTL). No new primary datasets were generated for this article and much of the general analysis/results are quite similar to their previous publication.

RESPONSE: We thank the reviewer for their constructive and helpful comments. The answers to each specific point are provided below, as well as the changes that have been made in the manuscript. Overall the manuscript has been substantially revised both in terms of writing and additional analyses (i.e. new results, updated figures and tables, and new supplementary figures), with the main conclusions unchanged but strengthened.

Major comments:

1. A major potential issue for this work is how to decouple the effects of gene expression and alternative splicing, and to control for biases that arise from this coupling. By definition, a gene non-expressed in the basal state but activated upon immune stimulation would never have a detectable sQTL signal in the basal state. Likewise, such a gene is more likely to have minor (and cryptic) splicing events detected in the activated state, given the increase in total gene expression upon immune stimulation. I worry that various statements and conclusions in this paper could be attributed to a bias for detecting regulated or cryptic splicing events when gene expression levels are higher (thus better power by RNA-seq). This issue needs to be examined and properly controlled for in a revised manuscript.

RESPONSE: We agree with the reviewer in that it is of major importance to properly consider the biases introduced by gene expression variation when studying alternative splicing (AS). Following the reviewer's comments, we have identified 3 situations where the coupling of splicing and gene expression could bias some of our results and interpretations:

- First, differences in the ability to detect AS events due to changes in gene expression could be mistaken for differential splicing upon stimulation. In our setting, this is avoided by focusing on genes that are highly expressed (FPKM > 10) both at the basal state and after stimulation, and considering only frequent splicing events (5% < PSI < 95%). Using these filters, and given the depth of our sequencing (35M reads per sample), we expect each kb of the minor transcript to be covered by ~15 reads per sample on average, as is now detailed in the Methods section (p26 ln10-15). Thus, the cases where we detect differential splicing upon stimulation are not likely to result from a lack of detection of the minor isoform in one of the conditions due to lower expression.

- Second, the apparent increase in the amount of noisy splicing observed after stimulation could be attributable to a better detection of cryptic splice sites due to higher gene expression after stimulation. To test for this possibility, we have now measured how the change in the number of cryptic splice sites (new Supplementary Fig. 4a) and noisy splicing (new Supplementary Fig. 4b) per gene varies according to changes in gene expression between the basal and stimulated state. While up-regulated genes present increased numbers of cryptic splice sites upon stimulation, we find that noisy splicing increases both among up- and down-regulated genes (despite down-regulated genes presenting a reduced number of cryptic splice sites after stimulation). Thus, the increase in noisy splicing that we report upon stimulation cannot be explained by a detection bias of cryptic splice sites in stimulated conditions. These results, and associated figures, have now been included in the section «Increased noisy splicing upon immune activation», p10 ln24 to p11 ln4.

- Third, to better consider the fact that sQTLs can only be detected in conditions where the gene is sufficiently expressed, we now used more conservative definitions to distinguish between (i) sQTLs of

genes expressed only upon stimulation, and (ii) sQTLs with a stimulation-specific effect (or response sQTLs). Specifically, for the latter group, we now focus only on sQTLs in genes that are sufficiently expressed at the basal state (FPKM > 10) and, moreover, present a significant difference in effect size before and after stimulation. In doing so, we have discarded 101 sQTLs that were not detected at the basal state due to insufficient power. All these results have been updated accordingly (see Results, p14 ln3-14, and Methods, p29 ln11-17).

We have also updated our definition of splice site activity (weak/alternative/constitutive) to make it more independent from overall gene expression. Namely, we now focus only on junctions that overlap genes that are expressed with FPKM > 1 (leading to a set of 1,106,965 splice sites) and assign splice sites as weak, alternative or constitutive based on the percentage of transcripts that use the splice sites rather than the absolute number of reads supporting the splice site. Following these changes, the first two sections of the Results have been updated and rewritten (p5 ln1 to p7 ln9), as well as the Methods section (p23 ln8 to p24 ln14).

2. The definition of condition-specific sQTLs may also need scrutiny and better rigor. We should not define condition-specific sQTLs simply as events detected as significant in one condition but not another: this could be due to statistical fluctuation or differential power in different conditions (e.g. the gene is more highly expressed in one condition so more RNA-seq coverage and better power for that condition). To define condition-specific sQTLs more rigorously we should look at effect size of genotypes on splicing in different conditions.

RESPONSE: As mentioned above and following the reviewer's suggestion, we now define as response sQTLs only those in genes that are sufficiently expressed in both conditions, and exhibit a significant difference in effect size between the basal and stimulated state. In addition, we now use a Bayesian model selection approach to assess the sharing of sQTLs across conditions. Specifically, for the 960 sQTLs where the corresponding genes are expressed at FPKM>10 across all 5 conditions, we now consider all 32 possible models of sQTL sharing (=2⁵ combinations of presence/absence across 5 conditions), and compute a likelihood for each possible sharing. For each model, we assume a constant effect size across conditions where the sQTL is present and set the effect size to 0 in conditions where the sQTL is absent. We then keep the model with the highest likelihood, to identify the most likely model of sQTL sharing across conditions. These new results are now presented in the section «Mapping the genetic control of context-specific splicing» p13 ln21 to p14 ln2, and supplemented by two updated figures (Fig. 4e and Supplementary Fig. 5b) and one note (Supplementary Note 4).

3. I have several concerns with the section on noisy splicing. Here, the authors define a metric for splicing noise and use this to quantify splicing noise across the five conditions. They quantify this value for all genes and conclude that noisy splicing increases after stimulation but is maintained at low levels for immune genes.

My main issue with the noisy splicing section is with the mathematical definition of noisy splicing. For all constitutive splice sites in a given gene, the proportion of correctly spliced reads are multiplied together and the complement of this product is used as a measure of noisy splicing. As a mathematical consequence of this definition, a gene with many introns would have noisier splicing than a gene with few introns. Indeed, the authors find a positive correlation between noisy splicing and intron number (page 10 line 6-7). However, the authors imply that this correlation is a biological phenomenon rather than a mathematical artifact.

It seems like the authors' rationale for multiplying the proportion of correctly spliced reads is to assume random assortment of noisy splicing and calculate the proportion of transcripts that have no incorrectly spliced introns. With this definition of noisy splicing, the values are not comparable across genes. However, the authors aggregate this measurement across all genes within the five conditions to come to their conclusions. Since the measurement is not comparable between different genes, and gene expression levels change across the five conditions, the authors' conclusions may be invalidated if they restrict their analysis individually within genes.

I would recommend that the authors restrict the noisy splicing analysis to the splice site level or pool the reads across all constitutive splice sites within a given gene to get an average

value and compare these values individually within genes across the five conditions to see if they come to the same conclusions. The latter part would be necessary to do an "apples to apples" comparison between different conditions.

RESPONSE: We thank the reviewer for this important comment. We understand the reviewer's concern that our definition of noisy splicing could hinder between-gene comparisons due to the strong positive correlation between intron number and noisy splicing. Our aim was to approximate the biological reality by estimating the percentage of transcripts that are non-functional, which is indeed « mathematically » increasing with the number of introns. However, measuring the mean rate of noisy splicing per splice site, as suggested by the reviewer, allows the measure to be comparable across genes. Using this metric, our results and conclusions supporting a global increase of noisy splicing upon stimulation remain unchanged. Furthermore, we show that this increase is also observed when comparing the rate of noisy splicing within a single gene, regardless of its fold change in expression upon stimulation (see new Supplementary Fig. 4b). These results are now presented in the corresponding section «Increased noisy splicing upon immune activation» (p10 ln5 to p11 ln9) and the associated figure has been accordingly updated (Fig. 3).

4. The authors compare putative regulatory variants that are shared between the sQTL and eQTL analysis and infer causality between SNP, expression, and splicing. However, readers may question the accuracy of these causal inferences, given the limitations of the data. It may help persuade readers to include specific examples of each type of causal model. For example, an sQTL in ERAP2 is known to introduce a premature termination codon and trigger nonsense mediated decay, so differential expression is causally impacted by differential splicing in this case.

RESPONSE: To address the reviewer's concerns, we have now extensively revised this section of the manuscript. Specifically, (i) we now explain in the main text the general principle of how causality inferences are performed (p12 ln17-20), as well as extend the Supplementary Note 3 (with a new Supplementary Figure 6) to provide an in-depth explanation of the underlying principles, and (ii) we now explicitly present the possible mechanisms through which splicing could regulate gene expression, or vice versa, as well as provide specific examples for each situation in the Results section (p12 ln12-16 and p12 ln24 to p13 ln8). This includes the case of *ERAP2* where our model correctly predicts splicing events leading to a nonsense transcript as causally affecting gene expression.

In addition, we have conducted new analyses to validate our causality inference, through the analysis of intronic reads that emanate from nascent, unspliced mRNAs (as in reference [6]). We reasoned that if gene expression is regulated through alternative splicing, we expect no change in transcription rate between the two alleles of an sQTL. Conversely, in the opposite situation, we expect to find a significant difference in the amount of pre-mRNA between the two alleles. Our analyses strongly support these predictions, by showing, for example, that transcription rate differs between the two alleles in 89% of cases where we predict that splicing is driven by gene expression (see p13 ln12-19).

All these results are now presented in the section «Disentangling the genetic regulation of isoform usage and gene expression».

5. For the section on regulatory elements and predicted impact on splicing, the four categories of regulatory elements seem to pertain more towards gene expression rather than splicing. RBP binding sites, RNA secondary structure, and conserved sites might be better candidates to look at.

RESPONSE: We thank the reviewer for this useful suggestion. We have now tested enrichments for a number of additional splicing regulatory elements, including RBP binding sites, donor/acceptor splice sites, splice site flanking region (3-10 bp within the intron), branch points (18-39 bp upstream of acceptor site, consistently with [12]), and conserved sites. This new analysis has now been added to the section «Characterising the genetic bases of splicing regulation» (p12 ln1-5) and in the new Supplementary Fig. 5a.

Minor comments:

Figure 1d: I would recommend reversing the order of the estimated age legend so that the order of the color scale is in the same orientation with the data.

RESPONSE: This has been done as requested.

*Additionally, the brackets indicate that three types of comparisons were done, but it is a bit confusing why those specific sets of comparisons (but not others) were done. Also, the legend defines a two asterisk (**) but there is no (**) in the plot.*

RESPONSE: To clarify the plot and avoid confusion, we are now comparing for each definition of immune genes (i.e. GO, fold-change, or both), the age of splice sites of immune genes to the age of splice sites genome-wide. We have also changed the code for asterisks and now use the following code : * : $p < 0.001$.

Figure 2c: It is a bit confusing what the brackets refer to and why there are only three sets of asterisks. Shouldn't there be four sets of comparisons (i.e. the four treatments to NS)?

RESPONSE: Following the reviewer's comment and to avoid any confusion, we have now removed significance from this figure. Significance is still reported in the text.

Figure 3b: The labels on the x-axis seems a bit strange. The last tick mark is labeled as ">20" but the preceding tick mark is "20". Should the last tick mark be changed to ">=30"?

RESPONSE: This figure has now been replaced by a figure showing the variation across genes in the rate of noisy splicing per site. The new figure 3b is now shown on log scale and the reading should be more intuitive.

Figure 3d: The NS plot should be added here for completeness.

RESPONSE: Following major changes in the structure of the manuscript, this analysis has been removed. As a consequence, we have removed Figure 3d as a whole.

Figure 4fg: These plots show the isoform expression levels, but a plot of PSI values would help emphasize the shift in alternative splicing (Similar to Figure 2e).

RESPONSE: Following the reviewer's suggestion, we now plot PSI values rather than isoform levels. In addition, gene FPKM levels are indicated through colour coding (grey squares) for each condition and genotype.

Figure 6b: The endpoints of intervals are not included in any of the marked intervals. i.e. if $\beta = 0.05$, is it included in the plot? Please modify this figure for accuracy.

RESPONSE: This has been modified according to the reviewer's suggestion.

Reviewer #3 (Remarks to the Author):

NCOMMS-18-18589: Defining the genetic and evolutionary architecture of alternative splicing in response to infection" by Prof Quintana-Murci and colleagues

Immune stimulation of monocytes is shown to increase mRNA isoform diversity through alternative splicing (AS). Perhaps surprisingly, a negative correlation between AS and gene expression is reported, suggesting that gene expression down-regulation correlates with an increase in AS. Finally, results are presented indicating that African- or European-specific alternative splicing patterns can be detected and that eight loci showing AS can be traced back to genetic Neanderthal admixture in Europeans.

This work is based on a RNAseq data set published in Cell in 2016 by the same research group (PMID:27768888) where AS was not considered. The research is an example of data re-analysis which may inspire others in the field. Surprising results emerged, namely the negative correlation of AS and RNA level reduction which may indicate hitherto unsuspected or not (yet) well documented relation between transcription frequency and splicing efficiency. The authors speculate that the increased detection of AS on lowly expressed genes may reflect a modulation of nonsense mediated mRNA decay (NMD). If this is the case, a stimulus-specific pattern of AS of lowly expressed genes would be expected. However I did not find this back in the manuscript? One alternative and radical interpretation a reader of this manuscript might come away with is that it may be possible that cells not only reduce the transcription of specific genes but also further inhibit specific gene expression levels by actively 'scrambling' RNAs through AS events. Hence the AS events qualified as 'noisy' by the authors could (highly speculatively) be an active gene expression downregulation mechanism. However, at this stage it is also possible that the increased AS detected at repressed genes is an artefact of the analysis pipelines. From the text going from P6L7 to P6L15, it is not obvious whether/how or not the analysis pipeline takes into account to what level an RNA is present (expressed as RPKM or FPKM) to call an AS. This could be explicitly stated if RNA level is not part of the filtering procedure.

AS as observed by the authors can be split into two categories: (i) AS that concerns RNAs that are well expressed and that will result in alternative protein isoforms being produced constitutively or under specific circumstances (unstimulated, LPS stimulation, virus exposure) and (ii) AS of (very) weakly expressed genes that show noisy splicing patterns. While the former is of general interest (human variation and evolution, immunology, RNA and protein research), the latter may reflect quixotic RNA metabolism or the coupling of splicing efficiency to transcription frequency, a field of investigation with many unresolved if not contentious questions. One weakness of the manuscript as it stands is that the authors do not distinguish well between (i) and (ii) in some sections of the results and discussion sections. This can be improved substantially, in particular at P9L20-22, p10L19-21, P14L21 (are these sQTLs associated with gene induction or gene repression?), P18L3 (does 'spliced' refer to AS, or to the ratio of intronic versus exonic reads?), P18L5 (are the 993 genes more or less alternatively spliced upon induction, or is there no quantitative bias in the isoforms produced and thus simply alternative AS? – as it stands this is a number but it does not really distinguish between (i) and (ii) above?)

RESPONSE: We thank the reviewer for their constructive and helpful comments. The answers to each specific point are provided below, as well as the changes that have been made in the manuscript. Overall the manuscript has been substantially revised both in terms of writing and additional analyses (i.e. new results, updated figures and tables, and new supplementary figures), with the main conclusions unchanged but strengthened.

Overall, the reviewer raises a number of points, to which we are responding here below:

- 1. How gene expression levels are taken into consideration to define splice site activity (weak/alternative/constitutive) should be clarified.** Indeed, RNA level was not part of the filtering procedure to define active splice sites. However, we have now updated our analyses and considered gene expression by focusing on junctions that overlap genes that are expressed with FPKM > 1 (leading to a set of 1,106,965 splice sites). Furthermore, we have assigned splice sites as weak, alternative or constitutive based on the percentage of transcripts that use the splice sites rather than the absolute number of reads supporting the splice site. Following these changes, the first two sections of the Results have been updated and rewritten (p5 ln1 to p7 ln9),

as well as the Methods section (p23 ln8 to p 24 ln14).

- The distinction between two types of AS should be clarified (i) functional splicing events that lead to a change in protein-coding isoform and (ii) noisy splicing events that result from random errors and do not lead to a functional protein.** To address this, we now evaluate the consequences of all 16,173 frequent AS events, based on the protein-coding potential of the resulting transcripts. In doing so, we find that ~56% of AS events result in a change in protein-coding isoform. The remainder are either AS events that concern non-coding isoforms (N=1,008, ~6%) or events where one of the possible resulting isoforms is non-coding (N=6,067, ~37%). Among these 6,067 events, the non-coding isoform was found to be, in most cases, expressed at low levels (see new Supplementary Fig. 2e), suggesting their rapid degradation by the NMD machinery. The distinction between these different types of AS events and their potential impact on protein function is now introduced throughout the manuscript. For example, we show that while there is general increase in non-coding transcripts after stimulation, when we focus on up-regulated genes, this trend is much less patent (see Results section «Understanding the nature of isoform changes in response to immune stimuli», p9 ln14-18, as well as the new Supplementary Figs. 3b and 3c).
- AS events qualified as ‘noisy’ could play an active role to downregulate gene expression.** The AS events that we describe as “noisy” in the section entitled «Increased noisy splicing upon immune activation» are indeed rare splicing events, which involve cryptic splice sites that are not evolutionary conserved. Thus, it is highly unlikely that such events would be beneficial to the organism and constitute an active mechanism of down-regulation. However, such a mechanism may exist among the 6,067 frequent AS events that alter the protein-coding potential of the gene. To address this question, we reasoned that frequent splice sites that are used to regulate gene expression through NMD are likely to be evolutionary conserved, whereas those that disrupt essential proteins will tend to be removed by natural selection. Focusing on exon skipping events for simplicity, we estimate that ~35% of frequent NMD-causing AS events involve evolutionary conserved splice sites and thus, might play an active role in down-regulating gene expression. This analysis is now included in the Results section (see p9 ln18 to p10 ln3 and Supplementary Fig. 3d).

Major comments:

1) It is known that neutrophil transcriptomes are rid with partially spliced RNAs (Orchestrated Intron Retention Regulates Normal Granulocyte Differentiation - PMID:23911323 (2013), Genetic Drivers of Epigenetic and Transcriptional Variation in Human Immune Cells - PMID:27863251 (2016)). It is also known that neutrophils are purified by CD14 beads (PMID:27863251). Usually the neutrophils are lost and monocytes are enriched upon selection for adhesion to Petri dishes. It is unclear to me whether the monocyte purification, conservation and stimulation protocols used (as published by the authors of the present study in Cell in 2016 PMID:27768888) preserved this neutrophil sub-population. Since a major part of the alternative splicing events concerns genes expressed at a low level (see comment xx), a concern is that the authors ‘scraped the bottom of the barrel’ in their search for alternative splicing events and ended up studying contaminating neutrophil RNAs rather than the monocyte RNAs they intended to study in the first place. One concrete way to check for neutrophil contamination would be to quantify the level of neutrophil-restricted/enriched RNAs such as FCGR3B and SLC44A2 in the 197 transcriptomes and in the AS events documented in the present manuscript. A second approach would be to perform a repeat of the major analyses of the paper exclusively on genes expressed better (> 10-fold?) by monocytes than neutrophils.

RESPONSE: The concern of the reviewer is that our splicing analysis on monocytes may be biased due to a neutrophil contamination. To explore this possibility, we have performed flow cytometry analysis using a panel of four antibodies (i.e. CD15, CD66b, CD16 and CD14) to characterise monocytes and neutrophil in (i) a purified CD14⁺ monocyte fraction obtained using the same experimental setting as in Quach et al. 2016, and (ii) the granulocyte/erythrocyte layer obtained from a whole blood density gradient centrifugation as a positive control for neutrophils. Neutrophils were defined as CD15⁺, CD66b⁺, CD16⁺ and CD14^{low}, whereas monocytes, which are CD15⁻ and CD66b⁻, are composed of CD14^{high}/CD16⁻ classical monocytes, the CD14^{high}/CD16⁺ intermediate monocytes and the CD14^{low}/CD16⁺ non-classical monocytes. This analysis shows that neutrophils are virtually absent from the purified CD14⁺ monocyte cell fraction (<0.12%), whereas they are, as expected, in

high proportions in the granulocyte/erythrocyte layer. This analysis is now presented as a new Supplementary Note 6 and Supplementary Fig. 8.

2) *Have the authors thought to deposit all the new alternative splice sites to a relevant RNA data base? Have the data been shared with Phantom? Alternatively, the Sequence Ontology may provide a repository for the splice sites identified here?*

RESPONSE: None of the analyses in the present manuscript involve the discovery of new splice sites, as we restricted our analyses to known splice sites that are present in the intron database to limit artefacts due to mapping errors. Note, however, that all sequence data has been deposited to EGA according to standard data sharing policies (accession number EGAS00001001895).

3) *It is possible to classify genes as a function of the exon/intron read signal, by mapping RNAseq reads to the genome, or only to cDNAs and computing a ratio of signal using the two approaches. I wonder whether the genes that are expressed at low levels and that show AS are also genes which belong to one end of this spectrum, namely with as many or even more intronic reads than exonic ones. This may help determine whether the source of variation in AS, as defined here, concerns nuclear RNAs that will most likely not give rise to proteins versus exported mature RNAs with a corresponding functional protein translation product.*

RESPONSE: We thank the reviewer for this suggestion. We now measure, for each gene, the total number of intronic reads to quantify the amount of nascent, unspliced mRNAs, as a proxy for the transcription rate. We then compare both transcription rate and global gene expression levels to the rate of noisy splicing per gene, and show that both metrics correlate negatively to the rate of noisy splicing (**figure I** for the reviewer, spearman $\rho < -0.19$, $p < 1.8 \times 10^{-74}$), suggesting coupling between transcription rate and splicing efficiency. However, variation in transcription rate does not completely account for the correlation between noisy splicing and gene expression, suggesting that noisy splicing also contributes to reduce expression levels through NMD. These analyses have now been included in the section entitled «Increased noisy splicing upon immune activation» (p10 ln 15-21).

Figure I. Relationship between noisy splicing, transcription and gene expression. 2D-density plots highlighting the correlation between log₁₀(rate of noisy splicing), gene FPKM (based on exonic reads, log transformed) and intronic RPKM (log transformed). Spearman correlations are reported for each comparison.

Minor comments:

Page7 Line12: the authors probably mean $|\log_2FC| > 1$ rather than $\log_2FC > 1$?

RESPONSE: Since our focus is immune genes, defined here as genes that are up-regulated upon immune stimulation, we consider indeed a $\log_2FC > 1$ and not the absolute fold change.

P10L21: “ due to a reduced rate of noisy splicing amongst highly expressed genes

RESPONSE: This sentence has been rephrased due to major changes in the paragraph relative to noisy splicing (see p11 ln6-9).

P11L4: “Most sQTLs (84%) were located within 10 kb of the AS event they control” could be toned down to ““Most sQTLs (84%) were located within 10 kb of the respective AS event, with 36% ...”

RESPONSE: This has been modified according to the reviewer’s suggestion (see p11 ln22-24).

P11L14: The authors mention “... when excluding the HLA region.” What proportion of the 36% of sQTLs falling directly within the boundaries of the AS event (P11L4) are concentrated in the HLA? How much of the 84% mentioned on P11L4 fall in the HLA region? One of the answers to these questions could be included in the text on P11L4.

RESPONSE: Defining *HLA* as the region of chromosome 6 spanning from 27.5 to 33.5 Mb, we find 22 genes with a sQTL in the *HLA* region, for a total of 38 sQTL out of 1464. Of these, 10 (26%) have their peak sQTL SNP falling directly within the boundaries of the AS event. We now directly report in the manuscript the fraction of sQTLs within 10kb of their respective AS event after exclusion of *HLA* genes (see p11 ln22-24).

P12L4-6: “These analyses indicate that the genetic determinants of isoform usage and gene expression are largely independent, yet, when they are not, splicing appears to be mainly driven by changes in gene expression” would be more accurate as “These analyses indicate that the genetic determinants of isoform usage and gene expression are largely independent, yet, when they are not, AS appears to be mainly driven by reduced gene expression”?

RESPONSE: The end of the sentence referred to the likelihood-based causal model selection analysis, that showed that among eQTLs that overlap sQTLs, variability in expression could account for the variability in splicing in 57% of the cases. As such, there is no evidence in the associated paragraph that AS is caused by a reduced gene expression. To avoid any confusion, we have now rephrased the sentence as follows: “Together, these analyses indicate that the genetic determinants of isoform usage and gene expression are largely independent; yet, when they are not, the regulation of AS through modulation of the transcription rate seems to be the predominant model” (see p13 ln16-19).

P13L9: what are ‘body measurements’? Can a literature reference to this trait be included? Is it body size/girth weight? Or ???

RESPONSE: « Body measurements » refers to a set of GWAS traits that were grouped according to the Experiment Factor Ontology (EFO) classification [13]. These mostly include anthropometric measurements such as height, BMI, waist hip ratio, height/weight/head circumference at birth, but also digit length ratio, and fat percentage. See <https://www.ebi.ac.uk/gwas/docs/file-downloads> for the full detail of GWAS-EFO mappings. This section has now been rephrased in the manuscript to clarify that the reported categories are based on EFO mappings, and provide examples from each EFO category (see p14 ln24 to p15 ln2)

P13L19: “Focussing on variants that ...” could be easier to catch if their number (16173??) were mentioned.

RESPONSE: Here we were referring to the 357 stimulation-sQTLs reported in the section «Mapping the genetic control of context-specific splicing». Following changes in the manner we define these stimulation specific sQTLs, we now report the updated number (235 sQTLs; 108+127) in this section (see p15 ln12-14).

P16L3: I would delete ‘finally’ from this sentence.

RESPONSE: This has been removed.

Colin Logie

References

1. Kornblihtt, A.R., et al., *Alternative splicing: a pivotal step between eukaryotic transcription and translation*. Nat Rev Mol Cell Biol, 2013. **14**(3): p. 153-65.
2. Naftelberg, S., et al., *Regulation of alternative splicing through coupling with transcription and chromatin structure*. Annu Rev Biochem, 2015. **84**: p. 165-98.
3. Schor, I.E., L.I. Gomez Acuna, and A.R. Kornblihtt, *Coupling between transcription and alternative splicing*. Cancer Treat Res, 2013. **158**: p. 1-24.
4. Saldi, T., et al., *Coupling of RNA Polymerase II Transcription Elongation with Pre-mRNA Splicing*. J Mol Biol, 2016. **428**(12): p. 2623-2635.
5. Bentley, D.L., *Coupling mRNA processing with transcription in time and space*. Nat Rev Genet, 2014. **15**(3): p. 163-75.
6. Gaidatzis, D., et al., *Analysis of intronic and exonic reads in RNA-seq data characterizes transcriptional and post-transcriptional regulation*. Nat Biotechnol, 2015. **33**(7): p. 722-9.
7. Li, Y.I., et al., *RNA splicing is a primary link between genetic variation and disease*. Science, 2016. **352**(6285): p. 600-4.
8. Raj, T., et al., *Integrative analyses of splicing in the aging brain: role in susceptibility to Alzheimer's Disease*. bioRxiv, 2017.
9. Wang, G.S. and T.A. Cooper, *Splicing in disease: disruption of the splicing code and the decoding machinery*. Nat Rev Genet, 2007. **8**(10): p. 749-61.
10. Alasoo, K., et al., *Genetic effects on promoter usage are highly context-specific and contribute to complex traits*. bioRxiv, 2018.
11. Ye, C.J., et al., *Genetic analysis of isoform usage in the human anti-viral response reveals influenza-specific regulation of ERAP2 transcripts under balancing selection*. bioRxiv, 2017.
12. Mercer, T.R., et al., *Genome-wide discovery of human splicing branchpoints*. Genome Res, 2015. **25**(2): p. 290-303.
13. Malone, J., et al., *Modeling sample variables with an Experimental Factor Ontology*. Bioinformatics, 2010. **26**(8): p. 1112-8.

REVIEWERS' COMMENTS:

Reviewer #1 (Remarks to the Author):

I appreciate the authors' thorough responses to my comments, as well as the insightful comments of the other reviewers. Overall, I am satisfied with the updated analyses and more measured conclusions presented in this revision. Thanks also for posting the code--though I wonder if it could be made more useful to the community by also posting (non-identifiable) intermediate files such as sQTL calls, etc. that served as input for generating figures.

One of the most important points raised by Reviewer 2 regarded the definition of condition-specific sQTL, which as previously defined could trivially arise due to differential power across conditions. The authors have rigorously addressed this question using a Bayesian model selection approach to distinguish tissue-sharing and tissue specificity.

I especially appreciate the extended description of various scenarios that may link gene expression and splicing and how these can be disentangled. The examples (e.g., of a nonsense isoform inducing NMD) were particularly helpful. The conclusion that NMD resulting from splicing alterations that generate nonsense isoforms can serve as a mechanism for modulating gene expression levels during immune response is very interesting.

There are a few sections that I still consider weaknesses, but in my opinion should not preclude publication:

1) I still find overlaps between sQTL and signatures of selection to be less convincing than other aspects of the paper, as it is based on an outliers with an albeit stringent, but arbitrary threshold. Moreover, several previous studies have demonstrated that the outliers detected by various selection scans (F_{st} , iHS , etc.) show very little overlap. The added requirement that the region be locally enriched for selection signatures makes the threshold more stringent than if this criterion was not included, but there is still no hypothesis test being conducted--just an overlap that is guaranteed to return results. I don't mean to say that the results are not interesting, but I stand by my previous assertion that this is very preliminary evidence.

2) Similarly, while the author's responses about the high LD threshold helped assuage my concerns, I still do not think that overlaps between many sQTL and many GWAS hits is sufficient to imply a causal relationship. Several groups have made important methodological contributions in this area (e.g., [https://www.cell.com/ajhg/fulltext/S0002-9297\(16\)30439-6](https://www.cell.com/ajhg/fulltext/S0002-9297(16)30439-6)), which could have provided more rigorous approaches to inferring causality.

Reviewer #2 (Remarks to the Author):

The authors have done a great job addressing my concerns and suggestions. I support the publication of this work.

Yi Xing

Reviewer #3 (Remarks to the Author):

The authors have satisfactorily clarified and extended their manuscript by smartly considering and incorporating almost every suggestion of the reviewers, including mine.

Colin Logie

REVIEWERS' COMMENTS:

Reviewer #1 (Remarks to the Author):

I appreciate the authors' thorough responses to my comments, as well as the insightful comments of the other reviewers. Overall, I am satisfied with the updated analyses and more measured conclusions presented in this revision. Thanks also for posting the code--though I wonder if it could be made more useful to the community by also posting (non-identifiable) intermediate files such as sQTL calls, etc. that served as input for generating figures. One of the most important points raised by Reviewer 2 regarded the definition of condition-specific sQTL, which as previously defined could trivially arise due to differential power across conditions. The authors have rigorously addressed this question using a Bayesian model selection approach to distinguish tissue-sharing and tissue specificity. I especially appreciate the extended description of various scenarios that may link gene expression and splicing and how these can be disentangled. The examples (e.g., of a nonsense isoform inducing NMD) were particularly helpful. The conclusion that NMD resulting from splicing alterations that generate nonsense isoforms can serve as a mechanism for modulating gene expression levels during immune response is very interesting.

RESPONSE: We thank the reviewer for their appreciation of our work and their support for publication, as well as for the suggestions that have helped improving the clarity of the manuscript.

There are a few sections that I still consider weaknesses, but in my opinion should not preclude publication:

- 1) I still find overlaps between sQTL and signatures of selection to be less convincing than other aspects of the paper, as it is based on an outliers with an albeit stringent, but arbitrary threshold. Moreover, several previous studies have demonstrated that the outliers detected by various selection scans (Fst, iHS, etc.) show very little overlap. The added requirement that the region be locally enriched for selection signatures makes the threshold more stringent than if this criterion was not included, but there is still no hypothesis test being conducted--just an overlap that is guaranteed to return results. I don't mean to say that the results are not interesting, but I stand by my previous assertion that this is very preliminary evidence.*

RESPONSE: We agree with the reviewer in that this evidence is somehow preliminary, although this is the best one can do when it comes to selection analyses. Following the reviewer's advice, we have now tuned down the discussion to acknowledge the uncertainty regarding the action of selection at the sQTLs we identify. Likewise, we mention the need for additional analyses based on simulations and/or ancient DNA to validate the presence of positive selection at the loci under consideration (p 21 ln 23 until p 22 ln 9).

- 2) Similarly, while the author's responses about the high LD threshold helped assuage my concerns, I still do not think that overlaps between many sQTL and many GWAS hits is sufficient to imply a causal relationship. Several groups have made important methodological contributions in this area (e.g., [https://www.cell.com/ajhg/fulltext/S0002-9297\(16\)30439-6](https://www.cell.com/ajhg/fulltext/S0002-9297(16)30439-6)), which could have provided more rigorous approaches to inferring causality.*

RESPONSE: We agree that co-localization tests constitute the gold standard for analyses of overlaps between GWAS loci and eQTLs/sQTLs. However, these methods require access to full summary statistics from the GWAS under consideration, and these data are often not publicly available. In particular, summary statistics were missing for several of the GWAS where we have the strongest evidence of co-localization with an sQTL (eg. Gastric cancer and *MUC1*; response to measles vaccine and *IFI44L*). Thus, while we believe that such colocalization analyses would help disentangle causal associations from simple co-occurrence of sQTL and GWAS loci, it is not possible to perform these analyses in a systematic manner and, in addition, we believe they are somehow beyond the scope of the current paper. Following the reviewer's comment, we have now nonetheless rephrased the discussion in a more cautious manner, and state that follow-up analyses of co-localization and Mendelian randomization are now needed to fully establish a causal role of splicing in disease. (p 22 ln 15-17).

Reviewer #2 (Remarks to the Author):

The authors have done a great job addressing my concerns and suggestions. I support the publication of this work.

Yi Xing

RESPONSE: We thank the reviewer for his appreciation of our work and his support for its publication. We also thank him for his previous suggestions that helped improving and strengthening the manuscript.

Reviewer #3 (Remarks to the Author):

The authors have satisfactorily clarified and extended their manuscript by smartly considering and incorporating almost every suggestion of the reviewers, including mine.

Colin Logie

RESPONSE: We thank the reviewer for his appreciation of our work and his support for its publication. We also thank him for his previous suggestions that helped improving and strengthening the manuscript.